# Reasons for East Siberia Winter Snow Water Equivalent Increase in the Recent Decades

Zhibiao Wang [1,2,3,*], Renguang Wu [4,5], Zhang Chen [3], Gang Huang [6] and Xianke Yang [1]

1 Center for Monsoon System Research, Institute of Atmospheric Physics, Chinese Academy of Sciences, Beijing 100029, China
2 State Key Laboratory of Severe Weather, Chinese Academy of Meteorological Sciences, Beijing 100081, China
3 Plateau Atmosphere and Environment Key Laboratory of Sichuan Province, School of Atmospheric Sciences, Chengdu University of Information Technology, Chengdu 610225, China
4 Key Laboratory of Geoscience Big Data and Deep Resource of Zhejiang Province, Department of Atmospheric Sciences, School of Earth Sciences, Zhejiang University, Hangzhou 310027, China
5 Southern Marine Science and Engineering Guangdong Laboratory (Zhuhai), Zhuhai 519082, China
6 State Key Laboratory of Numerical Modelling for Atmospheric Sciences and Geophysical Fluid Dynamics, Institute of Atmospheric Physics, Chinese Academy of Sciences, Beijing 100029, China
* Correspondence: wzb@mail.iap.ac.cn

**Abstract:** With the rapid warming in the past few decades, the snow water equivalent (SWE) in winter and spring decreased generally over the Northern Hemisphere, but an increasing trend occurred in some areas, especially in east Siberia. In this paper, we analyze the sources and reasons for the SWE increase in east Siberia in winter since 1979 and document projected future SWE changes in this region. The winter SWE changes in east Siberia were not significant over the past four decades until the 2000s, and the SWE increased rapidly thereafter. The SWE increase after the 2000s is mainly contributed by SWE in November, followed by that in winter, and attributed to the increase in snowfall. With the moisture budget diagnosis, we found that the atmospheric dynamic-induced moisture convergence (vertical motion effect and horizontal advection of moisture) are the reasons that contributed to the winter snowfall increase in east Siberia. As east Siberia is cold in winter, even under the high radiative forcing scenario, precipitation in east Siberia will continue to increase and be dominated by snowfall until the 2060s. Thereafter, with the rainfall increase and the accelerated snowmelt due to rising temperature, precipitation will gradually shift to rainfall type and the SWE may turn to decrease.

**Keywords:** SWE; precipitation increase; east Siberia; moisture budget; future change

## 1. Introduction

Snow cover is one of the fastest changing natural surface features in the Earth's climate system and highly persistent in cold seasons or regions [1]. Snow cover significantly affects atmospheric changes ranging from the synoptic scale to interannual or even longer time scales [2–7]. The albedo effect of snow is considered as a key element influencing the Earth's surface energy budget [8–10]. By reflecting downward solar radiation and cooling the overlying atmospheric layer, snow modulates local and downstream atmospheric circulation changes. In addition, the melting of snow in the warm seasons is an important source of freshwater on land in the Northern Hemisphere snow-covered areas [11], especially in arid and semi-arid mountain regions where snowmelt is a major factor in changes in river runoff and soil moisture [12,13]. For example, the increase in discharge corresponded to a decrease in snow cover during the snowmelt periods in the Siberian watersheds [14]. Moreover, earlier melting of snow due to rising air temperature also influences vegetation phenology and ecosystems and then affects the terrestrial carbon cycle [15].

Snow is sensitive to changes in air temperature and precipitation due to their modulation of snowfall and snowmelt [16,17], the two main factors that drive snow variations [18,19]. The distribution of snow cover varies largely due to spatiotemporal diversity in temperature and precipitation associated with high-frequency synoptic time-scale changes [20]. Under the significant warming over the past few decades, a dramatic decreasing trend in snow cover was observed in the majority of areas over the Northern Hemisphere in winter [21] and spring [14,17,22]. The long-term snow cover decline may be attributed to the atmospheric thermodynamical effect that induced a decrease in snowfall and an increase in snowmelt [18].

The changes in snow cover cannot reflect all the information of variations in snow, especially in thick snow areas [23]. The snow water equivalent (SWE) not only records the characteristics of spatial coverage of snow but also represents the depth of snow. So, different long-term trends of snow changes may appear in the same region based on snow cover and SWE. Recently, using the GlobSnow v3.0 SWE dataset, J. Pulliainen et al. [24] found that the SWE has a decreasing trend in the Northern Hemisphere over the past four decades in March, the month that has the maximum snow mass. However, the trend is not consistent across the Northern Hemisphere and an obvious increasing trend was observed in east Siberia for the period 1980–2018. This increase in SWE is considered to be mainly contributed by a marked increase in early winter snowfall in the last years of the period. However, the sources and reasons remain unclear for the SWE increase in east Siberia.

East Siberia is cold, with an average annual air temperature of about −10 °C, and the lowest air temperature can even reach −70 °C in winter [25]. The majority of precipitation falls as snow and snowmelt and is very little in cold seasons [26]. Thus, the winter SWE changes in east Siberia are closely related to snowfall. Nevertheless, the east Siberian region within the Arctic Circle has experienced a very significant temperature rise over the past few decades, even more than 0.7 °C/10a in winter, several times the rate of global average warming [27,28]. In addition, little is known about future changes in the cold season precipitation, snowfall, and rainfall in east Siberia.

This study documents the sources and reasons of the winter (DJF) SWE increase in east Siberia during the recent decades. A moisture budget is conducted to understand the mechanism of the SWE increase. Moreover, we evaluate the historical and future changes in precipitation, snowfall, and rainfall based on the sixth phase of the coupled model intercomparison project (CMIP6) model outputs.

## 2. Data and Methods

### 2.1. Study Area

East Siberia is located in the easternmost part of Eurasia (Figure 1a). The majority of the area is within the Arctic Circle. The annual average air temperature ranges from −10 °C to −15 °C, and the lowest and highest air temperatures are −70 °C and 30 °C [25]. The annual average precipitation ranges from 140 mm to 270 mm. In the cold season, precipitation is dominated by snowfall. The mean SWE has a remarkable month-to-month change. Widespread snow accumulation begins in October and reaches its maximum in the following March and melts quickly thereafter (Figure 1b).

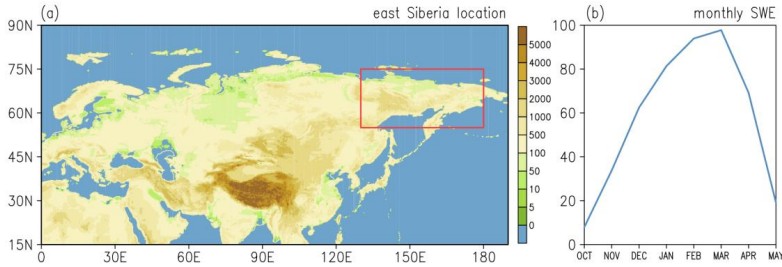

**Figure 1.** (**a**) Location of east Siberia (red box covered area) and (**b**) climatological monthly mean SWE (mm) in east Siberia from October to May. Shading in (**a**) denotes the elevation (m).

*2.2. Datasets*

　　To analyze the SWE changes over east Siberia in recent decades, we used the daily and monthly GlobSnow v3.0 Northern Hemisphere SWE dataset [29] (Table 1). The dataset was constructed by combining satellite-based passive microwave radiometer data with ground-based synoptic snow depth observations on the Northern Hemisphere terrestrial areas (non-high mountainous areas) and validated by independent in situ SWE data [29]. It is available on a 0.25° × 0.25° grid in daily and a 25 km × 25 km grid in monthly and spans the time period of January 1979–May 2018. We focused on the cold season (November to February) SWE changes. We converted the monthly data to regular 1° × 1° grids using the bilinear interpolation method from the Earth System Modeling Framework [30].

**Table 1.** List of observational and reanalysis datasets used in the study. The resolution and access site of each dataset are also listed.

| Dataset | Spatial Resolution | Time Resolution | Access Site |
|---|---|---|---|
| GlobSnow v3.0 | 0.25° × 0.25° 25 km × 25 km | Daily Monthly | https://globsnow.info/swe/archive_v3.0/ accessed on 3 August 2021 |
| ERA5 | 0.5° × 0.5° | Monthly | https://ecmwf.int/en/forecasts/dataset/ ecmwf-reanalysis-v5 accessed on 2 March 2022 |
| WFDE5 v2.1 | 0.5° × 0.5° | Hourly | https://cds.climate.copernicus.eu/cdsapp#!/ dataset/derived-near-surface- meteorological-variables accessed on 8 July 2022 |
| CRU ts4.06 | 0.5° × 0.5° | Monthly | https://crudata.uea.ac.uk/cru/data/hrg/ cru_ts_4.06/ accessed on 6 March 2022 |

　　To analyze the sources and reasons of the SWE changes, we used monthly cloud cover, radiation flux on single level and wind, vertical motion, and specific humidity on pressure levels from the ECMWF Reanalysis v5 (ERA5) [31] (Table 1). ERA5 is the newest generation ECMWF reanalysis, and it performs well in the high latitudes of the Northern Hemisphere [32,33].

　　We use hourly rainfall and snowfall generated using water, energy, and climate change (water and global change, WATCH) Forcing Data methodology applied to ERA5 (WFDE5) [34,35] (Table 1). This dataset is available on global land with a 0.5° × 0.5° horizontal resolution spanning the time period 1979–2019. We converted the quantities to the monthly mean for our analysis. Monthly surface air temperature and precipitation from the Climatic Research Unit (CRU) [36] of version 4.06 were used in the analysis (Table 1). The CRU dataset has a 0.5° × 0.5° spatial resolution for the period 1979–2019.

　　We used 18 model outputs from the CMIP6 [37] (Table 2) to evaluate the precipitation, snowfall, and rainfall projection. These include the model outputs during 1979–2100 with historical forcing until 2014 and projections under Shared Socioeconomic Pathways 5–8.5 (SSP585) scenarios during 2015–2100. We re-gridded all model outputs onto the 1.5° regular grid for analysis with the bilinear interpolation method from the Earth System Modeling Framework [30]. The ensemble mean of 18 model outputs was used to reduce the uncertainty of the model results. The historical simulations were compared with the observations to validate the reliability of model results.

**Table 2.** Climate model outputs in the CMIP6 historical and SSP585 simulations that were used in the study. The spatial resolution of each model is also listed.

| CMIP6 Models | Resolution (Lon × Lat) |
|---|---|
| ACCESS-CM2 | $1.875° \times 1.25°$ |
| BCC-CSM2-MR | $1.125° \times 1.125°$ |
| CanESM5-CanOE | $2.8125° \times 2.8125°$ |
| CNRM-CM6-1-HR | $0.5° \times 0.5°$ |
| CNRM-CM6-1 | $1.40625° \times 1.40625°$ |
| CNRM-ESM2-1 | $1.40625° \times 1.40625°$ |
| FGOALS-f3-L | $1.25° \times 1.0°$ |
| GFDL-ESM4 | $1.25° \times 1.0°$ |
| INM-CM4-8 | $2.0° \times 1.5°$ |
| INM-CM5-0 | $2.0° \times 1.5°$ |
| IPSL-CM6A-LR | $2.5° \times 1.268°$ |
| MCM-UA-1-0 | $3.75 \times 2.25°$ |
| MIROC-ES2L | $2.8125° \times 2.8125°$ |
| MIROC6 | $1.40625° \times 1.40625°$ |
| MPI-ESM1-2-LR | $1.875° \times 1.875°$ |
| MRI-ESM2-0 | $1.125° \times 1.125°$ |
| NorESM2-MM | $1.25° \times 0.9375°$ |
| UKESM1-0-LL | $1.875° \times 1.25°$ |

*2.3. Methods*

The moisture budget is widely used to analyze precipitation changes [38–41]. To understand mechanisms that contribute to snow mass increase in east Siberia, following Chou et al. (2009), the precipitation change is written as:

$$P' = - < \overline{\omega} \partial_P q' > - < \omega' \partial_P \overline{q} > - < V \cdot \nabla q >' + E' + residual, \qquad (1)$$

where $P$ is precipitation, $\omega$ is vertical velocity, $q$ is specific humidity, $V$ ($u$, $v$) is horizontal wind, and $E$ is evaporation. The contributions to precipitation changes include dynamic effect term $- < \omega' \partial_P \overline{q} >$, thermodynamic effect term $- < \overline{\omega} \partial_P q' >$, and horizontal advection terms $- < u \cdot \nabla q >'$, $- < v \cdot \nabla q >'$, respectively, related to atmospheric vertical motion, thermal conditions, and horizontal wind. The evaporation in cold seasons over the Northern Hemisphere high latitudes and the residual are both smaller terms and they may be omitted. Following Huang et al. [40], we simplified precipitation changes as follows:

$$P' = - < \overline{\omega_{500}} \partial_P q_{700}' > - < \omega_{500}' \partial_P \overline{q_{700}} > - < V_{700} \cdot \nabla q_{700} >', \qquad (2)$$

where $\omega 500$ is the vertical velocity at 500 hPa, $q700$ is the mean of specific humidity from 1000 hPa to 700 hPa, and V700 is the mean of horizontal wind from 1000 hPa to 700 hPa. The monthly data of variables are used in the above calculation.

We use the difference in the SWE between the end of a month and the beginning of a month as a measure of the change in the SWE in that month. In order to make the results more reliable, we use the average value of the last day of the current month and the first day of the next month as the value at the end of a month, and the average value of the first day of the current month and the last day of the previous month as the value at the beginning of a month.

This study focused on the trends of winter SWE change in east Siberia during 1979–2017. To make the trend analysis at the end of time period more reliable, we used atmospheric variables extended to 2019. Anomalies are calculated by the total values minus the climatology values in the time period. The low-frequency signal of variables was obtained by an 11-year low pass Gaussian filter. The statistical significance of trends was estimated based on the Mann–Kendall (MK) test [42,43].

## 3. Results

### 3.1. Feature and Sources of the SWE Increase

We noticed that the most significant increase in SWE in east Siberia is in winter for the period 1979–2017 except for the southeastern part (Figure 2a). From the time series of area-mean SWE in the region of 61°N–73°N, 133°E–175°E (blue box in Figure 2a), we found that the SWE change was small before the 2000s, but the SWE sharply increased after that (Figure 2c). The increasing trend of area-mean SWE in east Siberia is 4.4 mm/month/10a for the period 1979–2017 and 1.1 mm/month/10a and 15.0 mm/month/10a before and after the year 2000, respectively. The distribution of SWE trends for the period 2000–2017 is similar to that in the whole period but much larger and more significant (Figure 2a,b). The results indicated that the winter SWE in east Siberia increased rapidly after the 2000s.

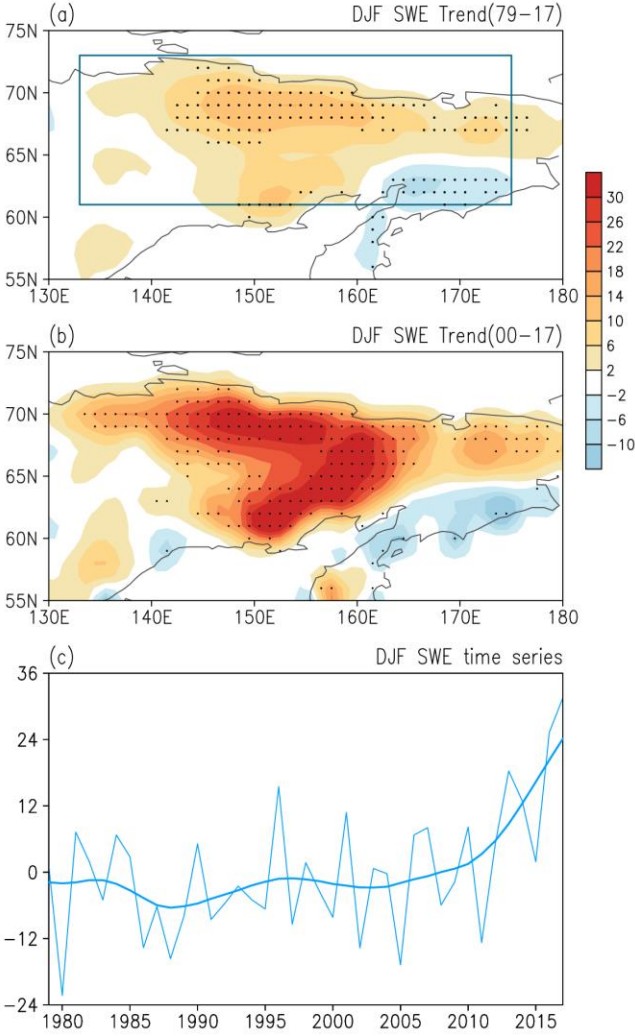

**Figure 2.** Linear trends of DJF SWE (mm month$^{-1}$ 10a$^{-1}$) for the period (**a**) 1979–2017 and (**b**) 2000–2017 based on monthly data. (**c**) Time series of area-mean DJF SWE (mm month$^{-1}$) anomalies for the period 1979–2017 in east Siberia (blue box covered area); thick line is obtained by an 11-year Gaussian filter. Dotted regions in (**a**,**b**) denote trends significant at the 95% confidence level.

The large extent of snow in east Siberia is mainly due to snowfall accumulation from October and later months. To understand the sources of winter SWE increase in east Siberia, we analyzed the trend of SWE difference for each month from October to February and calculated the area-mean SWE difference of each month during 1979–2017. In October, the trend of the SWE difference is sporadic and has no obvious spatial characteristics (figures not shown). The most significant increase in the SWE difference is

observed in November with a trend of 10.8 mm/month/10a after the year 2000 (Figure 3a,e). The increasing trends of the SWE difference are observed in the central and northwestern parts of east Siberia in December and January but in the central and eastern parts in February (Figure 3b–d). The trends of SWE difference in December to February are 6.6 mm/month/10a, 2.3 mm/month/10a, and 3.9 mm/month/10a, respectively, for the period 2000–2017. The results indicated that the snow mass changes from November to February (NDJF) have contributed to the increase in winter SWE in east Siberia after 2000, and the SWE in November has the largest contribution.

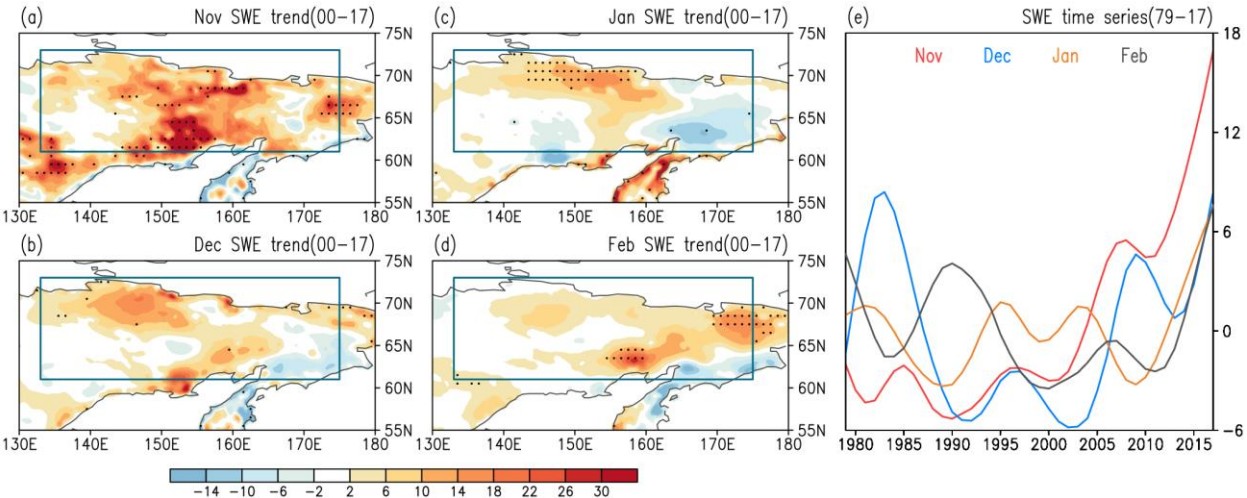

**Figure 3.** Linear trends of (**a**) November (Nov), (**b**) December (Dec), (**c**) January (Jan), and (**d**) February (Feb) SWE difference (mm month$^{-1}$ 10a$^{-1}$) for the period 2000–2017 based on daily data. (**e**) Time series of area-mean November (red line), December (blue line), January (orange line), and February (grey line) SWE difference (mm month$^{-1}$) anomalies for the period 1979–2017 in east Siberia (blue box covered area); all lines are obtained by an 11-year Gaussian filter. Dotted regions in (**a**–**d**) denote trends significant at the 95% confidence level.

### 3.2. Reasons for the SWE Increase

To understand the reasons for the SWE increasing trend in the recent two decades, we investigated the trends of surface air temperature and precipitation, the two factors that mainly contribute to snow changes. A significant temperature rising trend is observed in east Siberia, especially in the northern part (Figure 4a). From the time series of area-mean temperature, we found that the temperature changed slowly before the 2000s, but it sharply increased after that with the trend reaching 1.18 °C/10a for the period 2000–2019 (Figure 4c, orange line). At the same time, an obvious increase in precipitation exists in east Siberia, especially in the southern part (Figure 3b). The area-mean precipitation displays a slight decreasing trend during the period 1979–1999, and then a significant increasing trend after the 2000s (Figure 4c, blue line). The trend during the period 2000–2019 reaches 2.92 mm/month/10a.

The correspondence of the obvious SWE increase and significant temperature rise in east Siberia is inconsistent with the general opposite relationship between temperature and snow in the Northern Hemisphere cold seasons. This indicates that the temperature increase is not a direct cause of the SWE increasing trend. To illustrate the contribution of precipitation, we calculate the two quantities based on hourly snowfall and rainfall data. The trend and the area-mean changes of snowfall are highly consistent with those of precipitation, and the rainfall changes are very small (Figures 4 and 5). The results confirm that precipitation in east Siberia in NDJF was mainly in the form of snowfall. Due to the low temperature in this area, there is little snowmelt during NDJF. So, the variations of precipitation or snowfall are consistent with the SWE changes (Figures 2c and 4c blue line and 5c orange line).

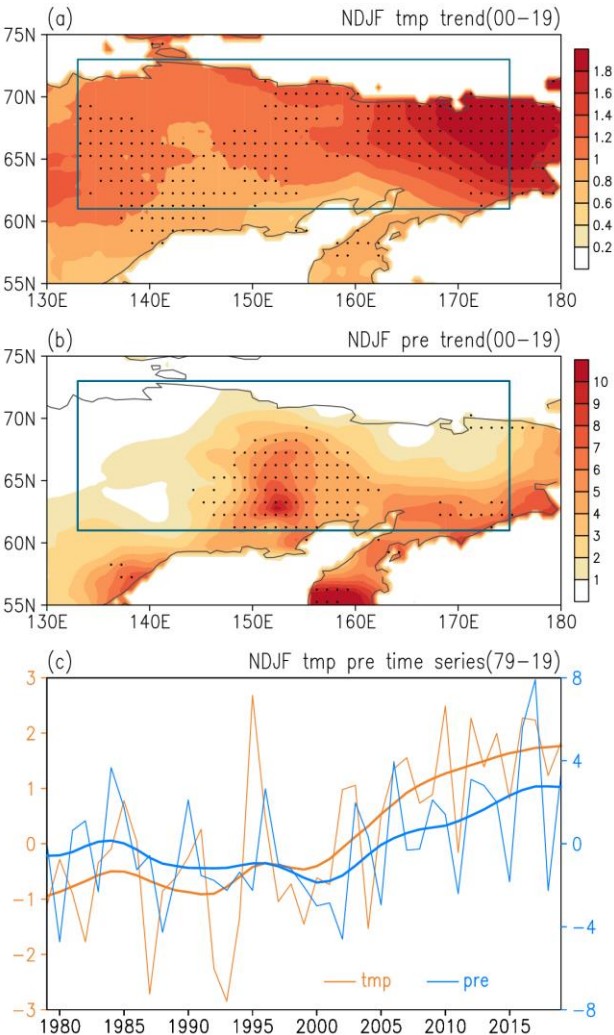

**Figure 4.** Linear trends of NDJF (**a**) temperature (tmp, °C 10a$^{-1}$) and (**b**) precipitation (pre, mm month$^{-1}$ 10a$^{-1}$) for the period 2000–2019. (**c**) Time series of area-mean NDJF temperature (orange line, °C) and precipitation (blueline, mm month$^{-1}$) anomalies for the period 1979–2019 in east Siberia (blue box covered area); thick lines are obtained by an 11-year Gaussian filter. Dotted regions in (**a**,**b**) denote trends significant at the 95% confidence level.

To illustrate the contribution of precipitation to the SWE changes, we analyzed the quantities associated with changes in precipitation. We noticed a significant increasing trend in upward motion of the atmosphere in central and eastern parts of east Siberia during the period 2000–2019 (Figure 6a), conducive to an increase in clouds in the region (Figure 6b). This result in a significant increase in downward longwave radiation (Figure 6c) that contributes to the surface air temperature rise. A rise in temperature is accompanied by an increase in the amount of water moisture in the atmosphere (Figure 6d). The increased moisture and enhanced upward motion are prone to produce more precipitation. However, during 1979–1999, there was a downward motion trend over most of the east Siberia region, accompanied by a decrease in middle cloud cover, downward longwave radiation, and specific humidity (Figure not shown).

To quantify the reasons of the increase in precipitation in east Siberia for the period 2000–2019, we conducted a moisture budget analysis to examine the contribution of atmospheric thermal and dynamical effects to the precipitation changes. From the simplified moisture budget equation (Equation (2)), we calculated the dynamic effect term $-<\omega'\partial_P\overline{q}>$ (−qdw), thermodynamic effect term $-<\overline{\omega}\partial_P q'>$ (−wdq), and horizontal

advection terms $- < u \cdot \nabla q >'$ (−udq), $- < v \cdot \nabla q >'$ (−vdq) based on monthly data. We found that the dynamic effect term has a significant positive contribution in the central and eastern parts of east Siberia and a negative contribution in the western part (Figure 7a), and the average contribution in the region is 1.08 mm/month/10a (Figure 7e). Moreover, the horizontal advection term also has an obvious positive contribution, especially in the southeastern part (Figure 7c,d), and the area means of $- < u \cdot \nabla q >'$ and $- < v \cdot \nabla q >'$ are 1.23 mm/month/10a and 0.61 mm/month/10a, respectively (Figure 7e). The contribution of the thermodynamic effect term is very small (Figure 7b). For the period 1979–1999, the negative dynamic effect term may be the reason for the weak decrease in the precipitation in the period (figure not shown).

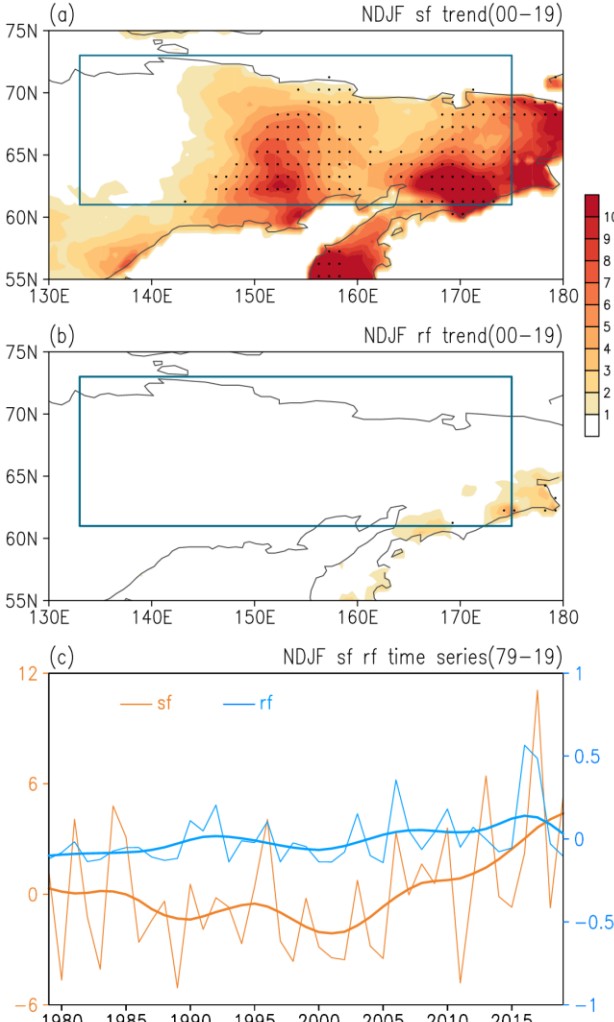

**Figure 5.** Linear trends of NDJF (**a**) snowfall (sf, mm month$^{-1}$ 10a$^{-1}$) and (**b**) rainfall (rf, mm month$^{-1}$ 10a$^{-1}$) for the period 2000–2019. (**c**) Time series of area-mean NDJF snowfall (orange line, mm month$^{-1}$) and rainfall (blueline, mm month$^{-1}$) anomalies for the period 1979–2019 in east Siberia (blue box covered area); thick lines are obtained by an 11-year Gaussian filter. Dotted regions in (**a**,**b**) denote trends significant at the 95% confidence level.

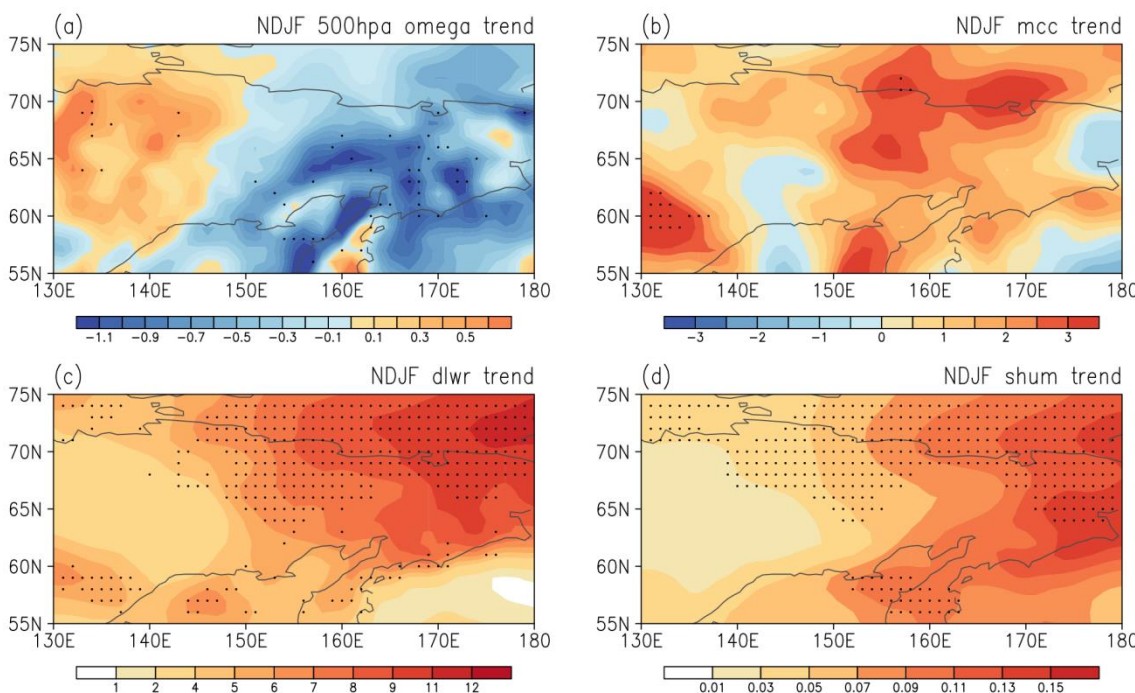

**Figure 6.** Linear trends of NDJF (**a**) omega at 500 hPa (Pa hour$^{-1}$ 10a$^{-1}$), (**b**) middle cloud cover (mcc, % 10a$^{-1}$), (**c**) downward longwave radiation (dlwr, W m$^{-2}$ 10a$^{-1}$), (**d**) average specific humidity from 1000 hPa to 700 hPa (shum, g kg$^{-1}$ 10a$^{-1}$) for the period 2000–2019. Dotted regions denote trends significant at the 95% confidence level.

### 3.3. Future Changes in CMIP6 Model Simulations

At present, the winter precipitation in east Siberia is still dominated by snowfall. Under continuous warming scenarios, will the situation continue and contribute to the SWE increase? To address this question, we evaluate the precipitation, snowfall, and rainfall that are closely related to the SWE changes in the historical forcing runs during 1979–2014 and the projections under SSP585 scenarios during 2015–2100 in NDJF in east Siberia based on output of 18 CMIP6 models (Table 2). Firstly, we calculated the trends of three variables in the historical forcing runs spanning 1979–2014 and 2000–2014 and the projections under SSP585 scenarios during 2015–2100 based on multi-model ensemble results, respectively. In the historical forcing runs for the period 1979–2014, the precipitation increased in east Siberia, especially in the southeastern part (Figure 8a). The increasing pattern is similar to the observations (figure not shown). The trend of area mean is 0.4 mm/month/10a for the period 2000–2014, lower than the observations. Moreover, the rainfall change is very small and the precipitation increase is mainly contributed by the snowfall (Figure 8b,c). For the period 2000–2014, the precipitation increased in central and northern parts of east Siberia, which is also mainly contributed by the snowfall (Figure 8d,e). This indicated that the precipitation increase in east Siberia was also dominated by snowfall in the historical forcing runs. In the projections under SSP585 scenarios, a significant increase in precipitation is observed in east Siberia for the period 2015–2100, especially in the eastern part of east Siberia (Figure 8g). Although the snowfall tends to increase in most continental areas, the increase in coastal regions is small or even there are decreases (Figure 8h). The precipitation type will shift gradually to rainfall under the continuous rise in temperature, especially in coastal regions (Figure 8i).

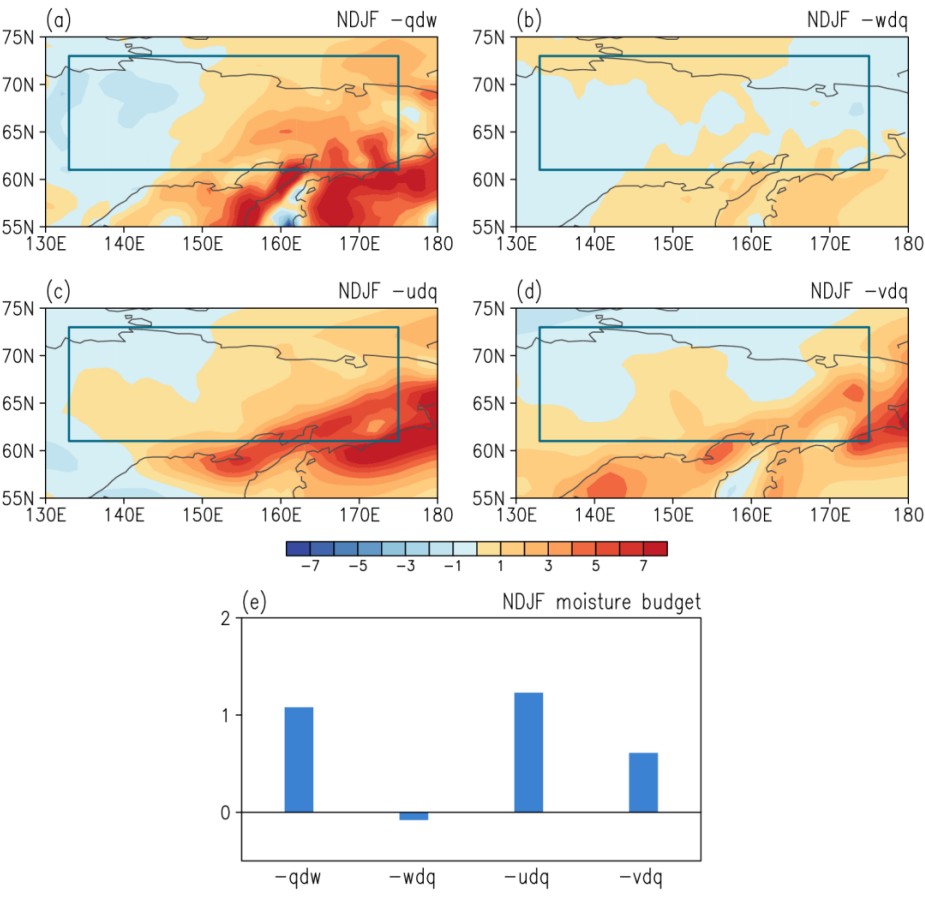

**Figure 7.** The moisture budget terms in NDJF of (**a**) dynamic effect ($-qdw$, mm month$^{-1}$ 10a$^{-1}$), (**b**) thermodynamic effect ($-wdq$, mm month$^{-1}$ 10a$^{-1}$), (**c**) zonal moisture horizontal advection ($-udq$, mm month$^{-1}$ 10a$^{-1}$), and (**d**) meridional moisture horizontal advection ($-vdq$, mm month$^{-1}$ 10a$^{-1}$). (**e**) Anomalies of moisture budget terms in NDJF of $-qdw$, $-wdq$, $-udq$, and $-vdq$ averaged over east Siberia for the period 2000–2019.

To clearly show the precipitation, snowfall, and rainfall changes in the historical forcing runs and the projections under SSP585 scenarios, we calculated the multi-model ensemble area-mean values of those variables from 1979 to 2100. The precipitation shows a continuously increasing trend through the period, especially after the 2060s when the changes are more significant (Figure 9). However, the trend of area-mean precipitation is higher before than after 2000, which may be due to the overly sensitive response of precipitation in the near-coastal regions to air temperature in the models. The results in Figure 8 suggest that the increase in precipitation after 2000 is obvious in most areas over northern parts of east Siberia, and certainly larger than that before 2000, but the increase is weak or even decrease is seen in the near-coastal regions. Snowfall also continues to increase in the whole period and is the dominant source of precipitation trend before the 2060s. However, after that, the rainfall increases significantly and part of the snowfall turns to rainfall (Figure 9). The results show that with the temperature rising in the projections under SSP585 scenarios, the precipitation in east Siberia in winter will continue to increase. However, even under the high forcing scenarios, the temperature in east Siberia is still very low until the 2060s, the increase in rainfall is slow, the precipitation is dominated by snowfall, and the SWE is likely to continue to increase due to snowfall increase and less snowmelt. After the 2060s, rising temperature accelerates the melting of snow and induces the gradual shift of precipitation to the rainfall type, and thus SWE may turn to decrease. The time series of area-mean NDJF precipitation, snowfall, and rainfall of each model are shown in Appendix A as Figure A1.

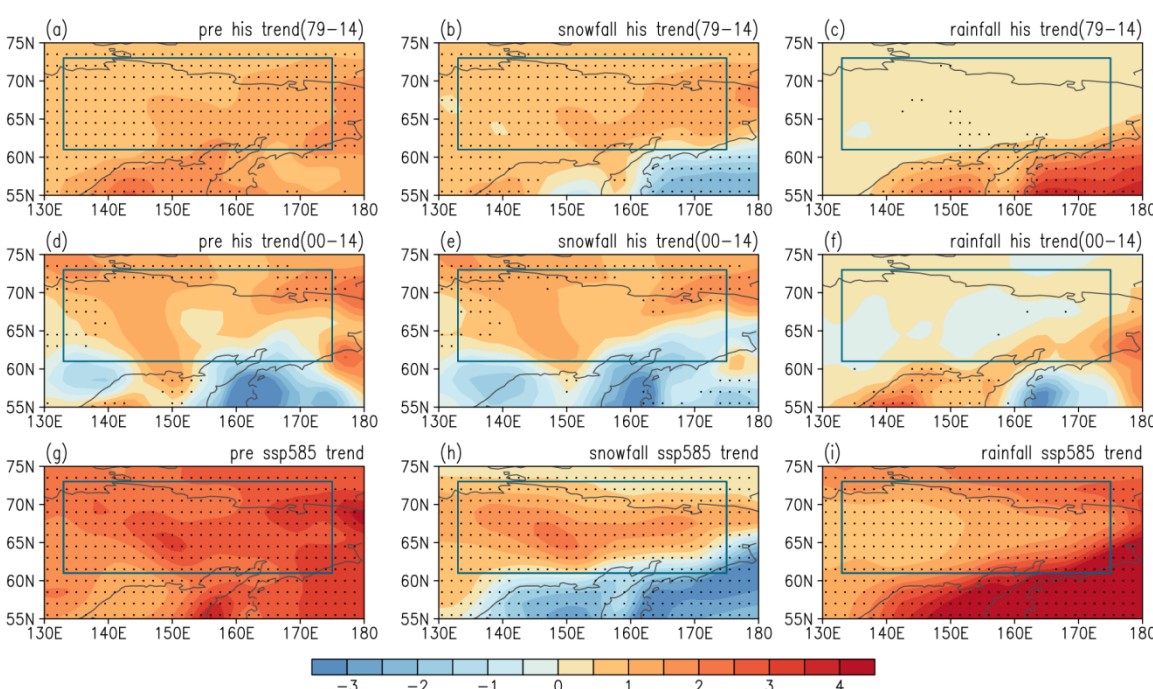

**Figure 8.** Linear trends of NDJF (**a,d,g**) precipitation (pre, mm month$^{-1}$ 10a$^{-1}$), (**b,e,h**) snowfall (mm month$^{-1}$ 10a$^{-1}$), and (**c,f,i**) rainfall (mm month$^{-1}$ 10a$^{-1}$) of (**a–c**) historical forcing for the period 1979–2014, (**d–f**) historical forcing for the period 2000–2014, and (**g–i**) projections under SSP585 scenarios for the period 2015–2100. Dotted regions denote trends significant at the 95% confidence level.

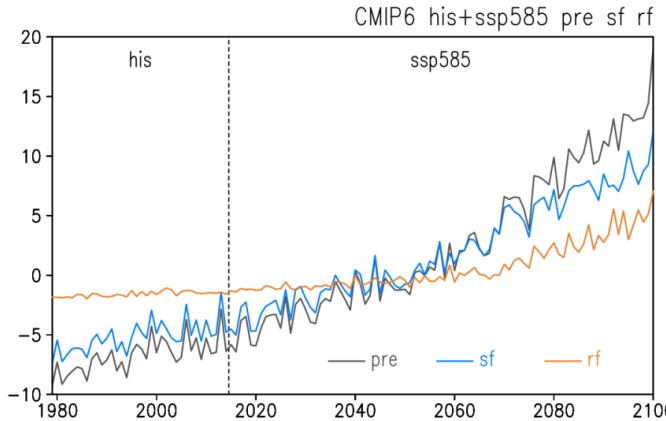

**Figure 9.** Time series of multi-model ensemble area-mean NDJF precipitation (grey line, mm month$^{-1}$), snowfall (blue line, mm month$^{-1}$), and rainfall (orange line, mm month$^{-1}$) anomalies for the period 1979–2100 with historical forcing until 2014 and projections under SSP585 scenarios during 2015–2100 in east Siberia (blue box covered area in Figure 8).

## 4. Discussion and Conclusions

In this study, we found that the winter SWE increase in east Siberia for the period 2000–2017 was mainly contributed by the SWE change in November, followed by that in the winter months. The snowfall changes are consistent with the precipitation changes. This indicated that the precipitation variations are mainly dominated by snowfall. The moisture budget analysis showed that the NDJF precipitation increase in east Siberia is influenced by the atmospheric dynamic effect and horizontal advection of moisture. This illustrated that the NDJF SWE changes in east Siberia after the 2000s are mainly contributed

by atmospheric vertical motion and horizontal winds, that is to say, the dynamic-induced moisture convergence is the main reason for the SWE changes.

The climate models simulated NDJF precipitation, rainfall, and snowfall trend pattern in east Siberia similar to the observations, but the values are smaller than the observations. Under the high forcing scenarios (SSP585), the precipitation will continue to increase for the period 2015–2100. However, unlike historical simulations in which the precipitation is dominated by snowfall, the rainfall in east Siberia will increase in the future, especially in coastal areas. After the 2060s, the precipitation will gradually shift to rainfall type. This implies a gradual expansion of the region with the temperature above zero degrees in winter in east Siberia, which will lead to accelerated snowmelt and may result in a decrease in SWE in the period after the 2060s.

The snow cover or SWE over the Northern Hemisphere mainly shows a declining trend. The Northern Hemisphere spring snow cover is projected to decrease by 8% $°C^{-1}$ in the coming 100 years relative to the 1995–2014 level [21]. Global warming is considered as one of the main causes [44]. For instance, the Northern Hemisphere middle latitude temperature change explains about 50% of the change in the Northern Hemisphere spring snow cover [17]. In addition to the atmospheric thermodynamic effects, the atmospheric internal variability has also contributed to the decline in March snow cover over Eurasia [17]. The present results show a new case that atmosphere dynamic-induced moisture convergence contributed to the increase in SWE in the recent two decades over east Siberia. The driver that contributed to internal circulation changes over east Siberia is a valuable question, which remains unclear and worthy of further study. Whether the contribution of this internal variability of the atmosphere associated with atmospheric dynamic effect to snow change will be sustained in the future is worth further investigation.

Observational analysis and model simulation suggested that Siberian snow has an influence on the Northern Hemisphere circulation changes via modulating the land–sea thermal contrast [45]. For example, the increase in snow in Siberia corresponded to the negative phase of the Arctic Oscillation and strengthened the Aleutian and Icelandic lows [46,47]. So, the increasing SWE in east Siberia may alter those relationships. Additionally, increased winter snow accumulation may raise the possibility of extreme flooding during snowmelt seasons [48]. Moreover, the newest climate model simulated snow mass changes have differences from the observations [49], especially for those changes contributed by atmosphere dynamics. The present results provide a basis for future simulation improvements.

**Author Contributions:** Conceptualization, Z.W. and R.W.; validation, Z.W., R.W. and G.H.; formal analysis and visualization, Z.W.; data curation, Z.W. and X.Y.; writing—original draft preparation, Z.W.; writing—review and editing, Z.W., R.W., G.H., Z.C. and X.Y.; supervision, R.W. All authors have read and agreed to the published version of the manuscript.

**Funding:** This study was funded by the National Natural Science Foundation of China Grants (42105028), the Second Tibetan Plateau Scientific Expedition and Research (STEP) (2019QZKK0102), the Open Research Fund Program of Plateau Atmosphere and Environment Key Laboratory of Sichuan Province (PAEKL-2022-K05), the National Natural Science Foundation of China Grants (41721004, 42141019, and 41831175), the Open Grants of the State Key Laboratory of Severe Weather (2022LASW-B23), the China Postdoctoral Science Foundation (2020T130640), and the Technological Innovation Capacity Enhancement Program of Chengdu University of Information Technology (KYQN202201).

**Data Availability Statement:** Not applicable.

**Conflicts of Interest:** The authors declare no conflict of interest.

**Appendix A**

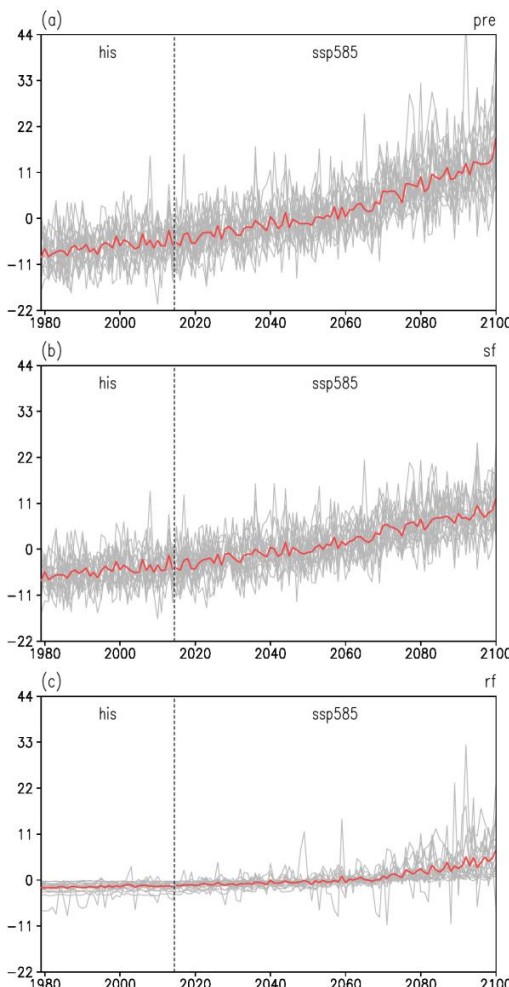

**Figure A1.** Time series of area-mean NDJF (**a**) precipitation (pre, mm month$^{-1}$), (**b**) snowfall (sf, mm month$^{-1}$), and (**c**) rainfall (rf, mm month$^{-1}$) anomalies of each model (gray line) and multi-model ensemble (red line) for the period 1979–2100 with historical forcing until 2014 and projections under SSP585 scenarios during 2015–2100 in east Siberia.

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
