# Peer review of "Reasons for East Siberia Winter Snow Water Equivalent Increase in the Recent Decades"

_remotesensing, doi:10.3390/rs15010134_

Round 1

Reviewer 1 Report

Review of “Reasons for east Siberia winter snow water equivalent increase in the recent decades” by Z. Wang, R. Wu, Z. Chen, G. Huang, and X. Yang

Using a suite of satellite data, in-situ observations, latest reanalysis (ERA5) and CMIP6 projections, the authors analyze the trends of winter snow water equivalent (SWE) over east Siberia in recent decades and in the coming century. It is revealed that the winter SWE over the east Siberia experiences significant increase from 1979, in particular after about 2000s, although other areas show steady decrease in response to the global warming. Along with the upward trends of the SWE, the surface temperature, precipitation (primarily snow fall), downward longwave radiation, middle-level vertical velocity, cloud cover and humidity also exhibit consistent upward trends. Further moisture budget analysis reveals that the upward trends of the precipitation primarily result from dynamic-induced moisture convergence and horizontal advection, so does the SWE. From the CMIP6 projections, it is found that the upward trends of the SWE likely will continue until 2060. After that, significant amount of precipitation will manifest as rainfall instead of snowfall. The associated hydrology (e.g., snowmelt and runoff increases) over east Siberia may change, thus changing the trends of the SWE. This manuscript is well written and easy to follow. After clarifying some major concerns, this paper may be accepted for publication.

Major Comments:

1, the reliability of the CMIP6 results is most troublesome. Figure 8 shows that the precipitation increases steadily from 1979 to 2014. The upward trends before 2000 are even stronger than that after, which is opposite to the observations. This result indicates that the CMIP6 models fail terribly to reproduce the observed trends over the target area, making the future projection questionable.

2, the causal claim (lines 216-219) “ a rise in temperature enhances the amount of water moisture in the atmosphere ….’ is questionable. Causal relationship can’t be established just based on data analysis. Well-designed numerical experiments with adequate models are needed to establish any causal claim.

3, it is suggested that the authors give deeper insights about why there is little (robust) trends before (after) 2000s.

Minor Comments:

1, it is suggested to revise the “dynamic effect” as “dynamic-induced moisture convergence” in the revised version. The horizonal advection is also “dynamic effect”!

2, line 31, although Cohen et al. (1991) claimed “the snow cover shows largest fluctuations …”, there is no quantitative evidence to support “the snow cover is the fast-changing surface feature”. How about high-latitude greening and Arctic sea-ice extent change? It is suggested to change the sentence to “ snow cover is one of fast-changing” unless there are robust evidence or literatures to support your claim.

3, line 86, signal => sigma

4, line 106, more details about the F-test used in this study should be given either in the footnote or in an Appendix.

5, it is suggested to use degree (instead of grid number) for CMIP6 model resolutions.

6, please clarifying the time scales represented with a top bar or a prime among lines 114-122 and other places in this manuscript.

7, please specifying which equation (line 122?) has been used to obtain Figure 6.

8, line 313, 200s => 2000s

Author Response

Review of “Reasons for east Siberia winter snow water equivalent increase in the recent decades” by Z. Wang, R. Wu, Z. Chen, G. Huang, and X. Yang

Using a suite of satellite data, in-situ observations, latest reanalysis (ERA5) and CMIP6 projections, the authors analyze the trends of winter snow water equivalent (SWE) over east Siberia in recent decades and in the coming century. It is revealed that the winter SWE over the east Siberia experiences significant increase from 1979, in particular after about 2000s, although other areas show steady decrease in response to the global warming. Along with the upward trends of the SWE, the surface temperature, precipitation (primarily snow fall), downward longwave radiation, middle-level vertical velocity, cloud cover and humidity also exhibit consistent upward trends. Further moisture budget analysis reveals that the upward trends of the precipitation primarily result from dynamic-induced moisture convergence and horizontal advection, so does the SWE. From the CMIP6 projections, it is found that the upward trends of the SWE likely will continue until 2060. After that, significant amount of precipitation will manifest as rainfall instead of snowfall. The associated hydrology (e.g., snowmelt and runoff increases) over east Siberia may change, thus changing the trends of the SWE. This manuscript is well written and easy to follow. After clarifying some major concerns, this paper may be accepted for publication.

Reply: Thanks very much for your Comments!

Major Comments:

1, the reliability of the CMIP6 results is most troublesome. Figure 8 shows that the precipitation increases steadily from 1979 to 2014. The upward trends before 2000 are even stronger than that after, which is opposite to the observations. This result indicates that the CMIP6 models fail terribly to reproduce the observed trends over the target area, making the future projection questionable.

Reply: Indeed, the results in Figure 9 (previous version is Figure 8) suggest that the area-mean precipitation increase trend is larger before than after 2000. This is likely due to the overly sensitive response of precipitation in the near-coastal regions to air temperature in the models. Figure 8 shows that the precipitation in the regions near the coast increased slightly or even decreased after 2000, but the increase of precipitation is obvious in most other regions over east Siberia, and certainly larger after than before 2000. Given the above feature, the CMIP6 results are not reliable in the regions near the coast, but agree with the observations in the inland regions.

In response, we add the following sentences to discuss this issue:

‘However, the trend of area-mean precipitation is higher before than after 2000, which may be due to the overly sensitive response of precipitation in the near-coastal regions to air temperature in the models. The results in Figure 7 suggest that the increase of precipitation after 2000 is obvious in most areas over northern parts of east Siberia and certainly larger than that before 2000, but the increase is weak or even decrease is seen in the near-coastal regions.’ (Lines 305-310)

2, the causal claim (lines 216-219) “a rise in temperature enhances the amount of water moisture in the atmosphere ….’ is questionable. Causal relationship can’t be established just based on data analysis. Well-designed numerical experiments with adequate models are needed to establish any causal claim.

Reply: As constrained by the Clausius-Clapeyron scaling, warming of the atmospheric column favors moistening. The water vapor holding capacity of the atmosphere increases by about 7% ℃-1. However, water vapor, as one of the greenhouse gases, may contribute to the temperature rise. Water vapor and its feedbacks play a role in the warming of the Arctic (including east Siberia). The processes such as albedo feedback play an important role in the Arctic warming. In addition, the water vapor in the high-latitude regions mainly comes from the warmer lower-latitude regions.

To make the description more appropriate, we modified the sentence to ‘a rise in temperature is accompanied by an increase in the amount of water moisture in the atmosphere ….’. (Lines 245-246)

3, it is suggested that the authors give deeper insights about why there is little (robust) trends before (after) 2000s.

Reply: Compared with the increased snowfall in east Siberia caused by the dynamic-induced water vapor convergence during the period 2000-2019, during 1979-1999, there was an obvious downward motion over most of the east Siberia region, accompanied by a decrease in middle cloud cover, downward longwave radiation, and specific humidity (Figure R1 a-d). With the moisture budget analysis, we found that this reduction is mainly contributed by the dynamical effect (Figure R1 e-i). So, the snowfall trend changes around 2000 is mainly attributed to the difference in dynamic-induced moisture convergence in the two periods. Why there is an obvious difference in atmospheric dynamics before and after 2000 is a question worthy of investigation in the future.

In response, we added sentence to describe the result. (Lines 248-250)

Figure R1. Linear trends of NDJF (a) omega at 500 hPa (Pa hour-1 10a-1), (b) middle cloud cover (mcc, % 10a-1), (c) downward longwave radiation (dlwr, W m-2 10a-1), (d) average specific humidity under 700 hPa (shum, g kg-1 10a-1) for the period 1979-1999. The moisture budget terms in NDJF of (e) dynamic effect (-qdw, mm month-1 10a-1), (f) thermodynamic effect (-wdq, mm month-1 10a-1), (g) zonal moisture horizontal advection (-udq, mm month-1 10a-1), and (h) meridional moisture horizontal advection (-vdq, mm month-1 10a-1). (i) Anomalies of moisture budget terms in NDJF of -qdw, -wdq, -udq, and -vdq averaged over east Siberia for the period 1979-1999. Dotted regions in (a-d) denote trends significant at the 95% confidence level.

Minor Comments:

1, it is suggested to revise the “dynamic effect” as “dynamic-induced moisture convergence” in the revised version. The horizonal advection is also “dynamic effect”!

Reply: Following the suggestion, we have modified the sentences. (Lines 23-24, and 333-334)

2, line 31, although Cohen et al. (1991) claimed “the snow cover shows largest fluctuations …”, there is no quantitative evidence to support “the snow cover is the fast-changing surface feature”. How about high-latitude greening and Arctic sea-ice extent change? It is suggested to change the sentence to “snow cover is one of fast-changing” unless there are robust evidence or literatures to support your claim.

Reply: Following the suggestion, we have modified the sentence. (Line 32)

3, line 86, signal => sigma

Reply: It should be “single”. The middle cloud cover and surface radiation flux are available only at one level. (Line 92)

4, line 106, more details about the F-test used in this study should be given either in the footnote or in an Appendix.

Reply:  The F-test is mainly used to detect mean state shifts without trend change (Wang 2000). In this paper, we need to test the significance of trend. It is not appropriate to use the F-test to examine the significance trend. Mann-Kendall (MK) test is usually used to examine the significance of trend (Mann 1945; Hamed 2008; Mondal et al. 2012). So, we redo the significance test and replaced the figures. We note that the difference between the results of the two test methods is small.

In response, we modified the sentence and figures. (Lines 142-143; Figures 2-6, and 8)

5, it is suggested to use degree (instead of grid number) for CMIP6 model resolutions.

Reply: Done.

6, please clarifying the time scales represented with a top bar or a prime among lines 114-122 and other places in this manuscript.

Reply: Done. (Lines 130-131 and 262)

7, please specifying which equation (line 122?) has been used to obtain Figure 6.

Reply: We have added specifics in the sentence. (Line 260)

8, line 313, 200s => 2000s

Reply: Done. (Line 332)

References:

Wang, X. L. Penalized maximal F test for detecting undocumented mean shift without trend change. Journal of Atmospheric and Oceanic Technology, 2008; 25(3), 368-384.

Mondal, A., Kundu, S., & Mukhopadhyay, A. Rainfall trend analysis by Mann-Kendall test: A case study of north-eastern part of Cuttack district, Orissa. International Journal of Geology, Earth and Environmental Sciences, 2012; 2(1), 70-78.

Mann, H. B. Nonparametric tests against trend. Econometrica: Journal of the econometric society, 1945; 245-259.

Hamed, K. H. Trend detection in hydrologic data: the Mann–Kendall trend test under the scaling hypothesis. Journal of hydrology, 2008; 349(3-4), 350-363.

Reviewer 2 Report

The manuscript, “Reasons for east Siberia winter snow water equivalent increase in the recent decades” is about why SWE has been increasing in eastern Siberia, especially since 2000.  Increasing precipitation, driven by dynamical effects and the horizontal advection of moisture, is the main driver of this increase.  While the area has also been warming as well, increasing precipitation is dominant since the region is cold enough for all precipitation to be snowfall during the winter (and presumably also cold enough for the snow not to melt).  The authors then use CMIP6 models to predict that SWE and snowfall is expected to increase in the region at least until the 2060s.

This study pretty convincingly shows that increased snowfall is responsible for increasing SWE amounts in east Siberia.  Although this is not necessarily new (e.g. increasing SWE is shown by GlobSnow3 and the region is cold enough to have stable snowpack all winter), it is interesting that the authors attribute this to increased dynamical forcing and horizontal moisture advection.  However, it is not clear why these things are happening.  While it may result in too many additions for this study, it should at least be discussed in the discussion section.  Another thing that could be discussed is what, if any, affects does this increased east Siberia snowfall have on surrounding regions (e.g. through modulation of the polar jet stream).  This would broaden the applicability of this study.

Another weakness is that while snowfall is treated very well, the impact of snow ablation is not.  For example, why wasn’t modeled SWE used or analyzed (either in ERA5 and the CMIP6 models), which would account for both snowfall and snow ablation.  In addition, looking at a simple ratio like end of winter SWE divided by accumulated snowfall would reveal how much mid-winter ablation there is.  While this may not be important in today’s climate since it is presumably cold enough to accumulate snow all winter, with climate change, this may shift.

Also, although most of the text is fairly clear, it could help to have an English speaker review the manuscript and make sure that grammar is correct, etc.

Specific Comments:

Line 22: Need to describe what the “dynamic effect” is here

Line 26: Figure 8 shows that rainfall starts to increase after 2060, though snowfall also increases after 2060, albeit at a slower rate.  Doesn’t this mean that SWE might still increase after 2060?

Line 53: This sentence is confusing and should be clarified

Section 2.1) Consider putting the links to the data sources in a table, or in the data availability statement at the end.

Line 90: Please define the acronym: WATCH.  Also, why do you have WFDE5 in parenthesis?

What is the data source for the CMIP6 models?

Line 117: It might be helpful to non atmospheric scientists to define what dynamic effect, thermodynamic effect, and horizontal advection mean.

Line 123: Does “under 700 hPa” mean “between the surface and 700 hPa”?

Line 138: These units are sort of strange.  For example, does 4.4 mm/month/10a mean for every 10 years, average SWE increases by 4 mm in all months, or does it mean for every 10 years, it increases by 4 mm in the first month, 4 more mm (=8mm) in the second month, and so on?

Line 194: This sentence is unclear and should be rephrased

Line 199: I don’t understand the calculation performed here.  Is it a simple snow model (e.g. simulating both the accumulation and ablation of snowfall. 

It would also be more convincing here to use a ratio like peak SWE divided by accumulated snowfall to show that despite the temperature increase, snowfall is still the primary determinant of how much snow is on the ground

Line 226: What could be driving changes in these terms.  For example, why is there more horizontal moisture advection into the region.  Is climate change a factor here.  Does it have to do with any other teleconnections?  Ideally, these things could be discussed in the last section of the paper.

Line 262: “Under the continuous temperature rising” – please rephrase

Line 284: Though snowfall is still increasing even after 2060.  How do we know that SWE won’t keep increasing despite the continued rainfall.

I think this section would benefit from more explicit analysis of snow ablation.  In the historical period, I believe you that it is too cold for there to be significant midwinter snowmelt, though in the future, this may not be the case.  Therefore, in addition to snowfall transitioning to rainfall around 2060, snow ablation affects might begin altering wintertime SWE distributions sometime in the future to.  It would be nice to know when that is. 

Do the CMIP6 models have a SWE variable that could be analyzed directly?

Line 313: “Vertical motion and horizontal winds” are not a compelling reason for increasing snowfall.  Could you expand on this (e.g. why the increased vertical motion and horizontal advection of moisture?)

Line 323: This last paragraph is hard to follow, and it feels out of place.  Could you talk about the computation of the trend values (and any nuances like what this paragraph seems to be about) in the Methods Section?

Also, please add a paragraph about broader implications (e.g. is it known, or speculated, how snowfall trends in eastern Siberia might affect climate variability elsewhere).  This would probably significantly enhance the applicability of this study.

Line 290: Usually it is “Discussion and Conclusion”

Author Response

The manuscript, “Reasons for east Siberia winter snow water equivalent increase in the recent decades” is about why SWE has been increasing in eastern Siberia, especially since 2000.  Increasing precipitation, driven by dynamical effects and the horizontal advection of moisture, is the main driver of this increase.  While the area has also been warming as well, increasing precipitation is dominant since the region is cold enough for all precipitation to be snowfall during the winter (and presumably also cold enough for the snow not to melt).  The authors then use CMIP6 models to predict that SWE and snowfall is expected to increase in the region at least until the 2060s.

This study pretty convincingly shows that increased snowfall is responsible for increasing SWE amounts in east Siberia.  Although this is not necessarily new (e.g. increasing SWE is shown by GlobSnow3 and the region is cold enough to have stable snowpack all winter), it is interesting that the authors attribute this to increased dynamical forcing and horizontal moisture advection.  However, it is not clear why these things are happening.  While it may result in too many additions for this study, it should at least be discussed in the discussion section.  Another thing that could be discussed is what, if any, affects does this increased east Siberia snowfall have on surrounding regions (e.g. through modulation of the polar jet stream).  This would broaden the applicability of this study.

Another weakness is that while snowfall is treated very well, the impact of snow ablation is not.  For example, why wasn’t modeled SWE used or analyzed (either in ERA5 and the CMIP6 models), which would account for both snowfall and snow ablation.  In addition, looking at a simple ratio like end of winter SWE divided by accumulated snowfall would reveal how much mid-winter ablation there is.  While this may not be important in today’s climate since it is presumably cold enough to accumulate snow all winter, with climate change, this may shift.

Also, although most of the text is fairly clear, it could help to have an English speaker review the manuscript and make sure that grammar is correct, etc.

Reply: Thanks very much for your comments!

Specific Comments:

Line 22: Need to describe what the “dynamic effect” is here

Reply: The dynamic effect refers to the effect associated with the atmosphere circulation changes.

In response, we modified the sentence. (Lines 23-24)

Line 26: Figure 8 shows that rainfall starts to increase after 2060, though snowfall also increases after 2060, albeit at a slower rate.  Doesn’t this mean that SWE might still increase after 2060?

Reply: Under the high radiative forcing scenario, both rainfall and snowfall increase after 2060s, but the rainfall increase is larger than the snowfall increase. This implies a gradual expansion of the region with the temperature above zero degree centigrade, which may lead to accelerated snowmelt and result in a decrease of SWE in the period after the 2060s.

In response, we modified the sentences to be more appropriate.

‘Thereafter, with the rainfall increase and the accelerated snowmelt due to rising temperature, precipitation will gradually shift to rainfall type and the SWE may turn to decrease.’ (Lines 26-28)

Line 53: This sentence is confusing and should be clarified

Reply: Variations in snow include changes in both extent and thickness. The snow cover mainly reflects the extent, but the SWE includes both extent and thickness.

Section 2.1) Consider putting the links to the data sources in a table, or in the data availability statement at the end.

Reply: Following the suggestion, we added a table listing the resolution and access site of the datasets. (Lines 103-104, and Table 1)

Line 90: Please define the acronym: WATCH.  Also, why do you have WFDE5 in parenthesis?

Reply: In response, we defined the WATCH and WFED5. (Lines 96-98)

What is the data source for the CMIP6 models?

Reply: The World Climate Research Programme (WCRP) Working Group on Coupled Modelling (WGCM) oversees the Coupled Model Intercomparison Project (CMIP), which is now in its 6th phase. About 112 climate models from 33 institutions around the world participated in CMIP6. In this study, we selected 18 climate models for analysis that provide historical simulations of snowfall and rainfall as well as future projections

Line 117: It might be helpful to non atmospheric scientists to define what dynamic effect, thermodynamic effect, and horizontal advection mean.

Reply: The dynamic effect refers to that associated with the atmosphere circulation changes, for example, atmosphere vertical motion changes. The thermodynamic effect refers to that attributed to atmospheric thermal conditions, for example, atmospheric moisture changes. Meteorologists describe the horizontal movement of variables (like moisture) by the wind as horizontal advection.

In response, we added specific words of description. (Lines 124-125)

Line 123: Does “under 700 hPa” mean “between the surface and 700 hPa”?

Reply: It should from 1000 hPa to 700 hPa.

In response, we modified the sentence. (Lines 130-131)

Line 138: These units are sort of strange.  For example, does 4.4 mm/month/10a mean for every 10 years, average SWE increases by 4 mm in all months, or does it mean for every 10 years, it increases by 4 mm in the first month, 4 more mm (=8mm) in the second month, and so on?

Reply: The monthly SWE unit is mm/month. The unit of trends should be mm/month/10a, which means that monthly averaged SWE increases 4 mm in every 10 years.

Line 194: This sentence is unclear and should be rephrased

Reply: The sentence is not necessary. We deleted it. (Line 225)

Line 199: I don’t understand the calculation performed here. Is it a simple snow model (e.g. simulating both the accumulation and ablation of snowfall. 

Reply: The hourly rainfall and snowfall are generated using water, energy and climate change (WATer and global CHange, WATCH) Forcing Data methodology applied to ERA5 reanalysis. Data have been adjusted using an elevation correction and monthly-scale bias corrections based on Climatic Research Unit (CRU) data (for temperature, diurnal temperature range, cloud-cover, wet days number and precipitation fields) and Global Precipitation Climatology Centre (GPCC) data (for precipitation fields only). Additional corrections are included for varying atmospheric aerosol-loading and separate precipitation gauge observations. The rainfall and snowfall are not generated via a simple snow model (Weedon et al. 2011; Weedon et al. 2014).

It would also be more convincing here to use a ratio like peak SWE divided by accumulated snowfall to show that despite the temperature increase, snowfall is still the primary determinant of how much snow is on the ground

Reply: Such a ratio may become very large in snow accumulated snowfall cases, leading to large uncertainty.

Line 226: What could be driving changes in these terms.  For example, why is there more horizontal moisture advection into the region.  Is climate change a factor here.  Does it have to do with any other teleconnections?  Ideally, these things could be discussed in the last section of the paper.

Reply: Indeed, the driver that contributed to circulation changes over east Siberia is a valuable question, which remains unclear and is worthy of further study.

In response, we added sentences in the last section to discuss this issue. (Lines 354-356)

Line 262: “Under the continuous temperature rising” – please rephrase

Reply: Following the suggestion, we modified the sentence. (Line 295)

Line 284: Though snowfall is still increasing even after 2060.  How do we know that SWE won’t keep increasing despite the continued rainfall.

Reply: The rainfall and snowfall are both increasing after the 2060s, and the value of rainfall trend is larger than the snowfall trend. This implies that the region with the temperature above zero degree centigrade has increased obviously, leading to rainfall increase and the snowmelt accelerated significantly. The SWE is determined by both snowfall and snowmelt. With the increase above zero degree centigrade, the snow will melt very quickly. This may contribute to the SWE decrease at some stage thereafter.

I think this section would benefit from more explicit analysis of snow ablation. In the historical period, I believe you that it is too cold for there to be significant midwinter snowmelt, though in the future, this may not be the case.  Therefore, in addition to snowfall transitioning to rainfall around 2060, snow ablation affects might begin altering wintertime SWE distributions sometime in the future to.  It would be nice to know when that is. 

Reply: Thanks for your very great suggestion! Indeed, it will be more convincing if we use snowmelt or directly use SWE from the climate models to project east Siberia wintertime SWE changes. Unfortunately, the majority of climate model projections do not provide the two variables. In addition, the model produced snow variables may include uncertainty. Therefore, the possible change of SWE can only be inferred indirectly through the change of rainfall and snowfall.

Do the CMIP6 models have a SWE variable that could be analyzed directly?

Reply: Only a few models have the SWE variable. Due to the small number of models, the results of the analysis will not be representative, so this study did not conduct the analysis of model snow quantities.

Line 313: “Vertical motion and horizontal winds” are not a compelling reason for increasing snowfall.  Could you expand on this (e.g. why the increased vertical motion and horizontal advection of moisture?)

Reply: Vertical motion and horizontal winds of the atmosphere are circulation changes. Horizonal wind change induce moisture convergence changes, leading to moisture changes in the atmospheric column. The vertical motion changes alter the condensation of water vapor in the atmosphere so that snow may form and fall, So, analyzing the factors that contributed to the circulation changes is a valuable issue. Due to the length limitation of the paper, we just discussed it in the last section. (Lines 354-356)

Line 323: This last paragraph is hard to follow, and it feels out of place.  Could you talk about the computation of the trend values (and any nuances like what this paragraph seems to be about) in the Methods Section?

Reply: In the previous vision, we discussed the uncertainty of SWE increase trends based on different time resolution datasets. In the present version, we move this part to the last paragraph of section 3.1 to explain the possible reasons that caused the value difference. (Lines 200-206)

Also, please add a paragraph about broader implications (e.g. is it known, or speculated, how snowfall trends in eastern Siberia might affect climate variability elsewhere).  This would probably significantly enhance the applicability of this study.

Reply: Following the suggestion, we added a paragraph in the last section to discuss the implications:

‘Observational analysis and model simulation suggested that Siberia snow have an influence on the Northern Hemisphere circulation changes via modulating the land-sea thermal contrast [47]. For example, the increase of snow in Siberia corresponded to negative phase of the Arctic Oscillation and strengthened the Aleutian and Icelandic lows [48, 49]. So, the increasing SWE in east Siberia may alter those relationships. Besides, increased winter snow accumulation may raise the possibility of extreme flooding during snowmelt seasons [50]. Moreover, the newest climate model simulated snow mass changes have differences from the observations [51], especially for those changes contributed by atmosphere dynamics. The present results provide a basis for future simulation improvements.’. (Lines 359-368)

Line 290: Usually it is “Discussion and Conclusion”

Reply: In the last section, we first summarized and then discussed the valuable issues worth analyzing in the future and the implications of this study.

References:

Weedon, G. P., Gomes, S., Viterbo, P., Shuttleworth, W. J., Blyth, E., Österle, H., ... & Best, M. Creation of the WATCH forcing data and its use to assess global and regional reference crop evaporation over land during the twentieth century. Journal of Hydrometeorology, 2011; 12(5), 823-848.

Weedon, G. P., Balsamo, G., Bellouin, N., Gomes, S., Best, M. J., & Viterbo, P. The WFDEI meteorological forcing data set: WATCH Forcing Data methodology applied to ERA‐Interim reanalysis data. Water Resources Research, 2014; 50(9), 7505-7514.

Reviewer 3 Report

Comments on “Reasons for east Siberia winter snow water equivalent increase in the recent decades” by Wang et al. submitted to Remote Sensing

General comments

I have now read the paper “Reasons for east Siberia winter snow water equivalent increase in the recent decades”. The authors present a paper that aims at analysing the reasons of the winter (DJF) SWE increase since the year 1979 in east Siberia. For this region, the authors also provide projections of future SWE changes. They found that significant SWE increases occurred in the period 2000-2017, mainly in November (followed by the other winter months). The authors attributed this occurrence to NDJF precipitation increases due to the atmospheric dynamic effect and horizontal advection of moisture. Finally, through climate simulations, under high forcing scenarios, the authors showed that precipitation will continue to increase until 2100, while after the 2060s the precipitation could shift to rainfall type.

In general, the paper has clear and interesting aims. In addition, the reported findings are of great interest for the scientific community. Finally, the theme fits well into the scope of the journal Remote Sensing. However, there are several aspects that need to be improved/clarified before recommending this paper for publication.

Main comments

Data and Methods

Currently, the description of your study area is completely missing. I understand that it is the east Siberia, but please consider that not all readers (including myself) are well informed about the characteristics of the region. Therefore, my suggestion is to add a Study area section to the Data and Methods where you could provide geographical and climatic information. In addition, can you provide here more information about estimated monthly SWE means in your study region? I think it would be nice to have a very rough idea about the seasonal SWE evolution in the analysed region (maybe there are even some observational data…). In this section, I also recommend to add a figure (it will be the Figure 1) in which you show the location of your study area in the world (or in the Northern Hemisphere, if you prefer), its borders (even roughly, like you did in Fig. 1a), etc.

Datasets

Although I am not an expert in using all the datasets you used in this paper, I feel that several bits of information are missing here. In addition, I like the idea to keep the section short (as well as the entire paper), but I think this is too much.

(i) I would recommend to clearly explain why you chose some datasets and to do what. At the moment, all possible explanations are up to the reader, since all your datasets “pop-up” in a menu-style list without further explanations. Please, “take the reader in the hand” and try to explain why and how you are doing things; you may want to do this also by describing all used datasets (and variables) in a table.

(ii) In this study, you completely rely on gridded remote sensed and reanalysis data, which is not necessarily bad. However, have you considered validating some of the used products with (potentially) available observational data? Do you have some ideas about the estimated errors and uncertainties associated to the products you used in your study area? Please, try to be very precise and rigorous since this is necessary to allow the reader assessing the overall quality of your approach and the uncertainties in your results.

(iii) Have you assessed the uncertainties in your scenarios? Currently, I feel that too many “certainties” about your outcomes and predictions are conveyed by your paper. For instance, this is what you wrote in the abstract “As east Siberia is cold in winter, even under the high radiative forcing scenario, precipitation in east Siberia will continue to increase and is almost dominated by snowfall until the 2060s. Thereafter, the precipitation will shift to rainfall type…”. I suggest being more cautious (including your writing style), also considering that a shift from solid to liquid precipitation is not trivial to model and predict; great uncertainties might exist in this. You might want to explain how you quantified the uncertainties here or, better, in the following sub-chapter (Methods).

RESULTS

The Results section is an amalgam of results, interpretation, discussion, and methodology. Lines 132-134: this is discussion. Lines 159-166, this part should be in the methodological section (it is relevant for your work). Are there previous studies that applied this approach? Can you provide some references? Lines 213-219: mixture of results and interpretation. Lines 238-240, 261-264, and 281-284: this is interpretation (further details below).

CONLUSIONS AND DISCUSSION

I think that this part should be completely rewritten.

(i) Currently, this section does not contain a discussion nor a conclusion. Essentially, you somehow repeat what you already wrote in the previous section (except the last paragraph). I suggest describing your results and interpreting them (you did it, already), second, you should discuss them in the context of previous research. Merging and rewriting the current Results section with the current Conclusions and Discussion might help you in doing this, obtaining a new section: Results and Discussion.

(ii) At the moment, your paper heavily relies on the interpretation rather than discussion of data. I understand that this is not an extremely specific comment, but I would suggest you to try improving your discussion with this in your mind. For instance, are there other regions/area in the world with increasing SWE in the recent decades/years? Are there papers discussing about these regions/areas? Any analogies with your case? What are the implications of your findings? Do you have remaining questions, doubts, new perspectives, etc.? These are just some suggestions.

(iii) Please, write your conclusions in a separated section. Currently, I cannot find a real conclusion in your paper. The Conclusions should summarise the conducted research both with respect to its place among the published works and in terms of future perspectives. Also, what did we learn from your study? Why is your study relevant for the scientific community? Therefore, in order to better convey the main findings of the paper, it would be better to write few, brief conclusive highlights and then adding some perspectives.

(iv) In the last paragraph you discuss some uncertainties regarding your approach. First, I do not think that this paragraph should be at the end of your paper, in the Conclusions. The take-home message for the reader seems to be this one, which is a methodological issue, essentially. I think you have way more relevant take-home messages and potential future research perspectives to explore. Second, this part could go into the new Results and Discussion section, maybe in a sub-chapter called Research limitations and uncertainties, in which you might also want to discuss the uncertainties related to your research (please, see previous comments about this issue).

FIGURES

I find the resolution of the figures to be pretty low, which could be due to an automatic downsampling performed by the submission system (I do not know this). If this is not the case, I warmly recommend improving the resolution of all figures.

Detailed comments

Keywords: (1) “Snow water equivalent” - I would use “SWE” since you already used “snow water equivalent” in the title. (2) “Sources and reasons of increase” - This sounds like a weird keyword. I would use other keywords such as “precipitation increase”, “climate change”, “snowfall increase”, etc. These are just some hints.

Line 25: “…and is almost dominated by snowfall until the 2060s”. How can something be “almost” dominated by something else? Can you rephrase this?

Line 26: “Thereafter, the precipitation will shift to rainfall type”. I would be more cautious; properly communicating scientific uncertainty is vital, especially when this is about climatic scenarios and potential precipitation phase changes. In addition, from what you write, one could think that all solid precipitation will shift to liquid, which is not the case. Please, try to be more precise since this is a very strong statement, which is not even supported by your results. Indeed, there might be an increase in rainfall, although snowfall is still predicted to be predominant in most areas.

Line 31-32: “Snow cover is the fastest-changing natural surface feature in the Earth’s climate system…”. What do you mean? Maybe you meant cryosphere instead of Earth’s climate system?

Line 36: “…local and downstream atmospheric circulation changes”. Are you talking about mountains, here?

Lines 41-41: “For example, about 50% of runoff in western Himalayan watersheds comes from seasonal snowmelt [14]”. I do not think this sentence is pertinent to your study. It would be better to focus on geographical settings (and environmental processes) more relevant for your study region (and similar regions).

Lines 65-66: “East Siberia is cold, especially in winter when the majority of precipitation falls as snow and snowmelt is very little in cold seasons”. How cold? Can you provide more climatic information? Here or in the Study area section.

Lines 72-73: “This study documents the sources and reasons of the winter (DJF) SWE increase in east Siberia during 1979-2018”. Here, you state that the investigated period is 1979-2018. However, the actual period you analysed for SWE trends is 1979-2017, right? In addition, for precipitation (and other climatic data) it seems that you used the 1979-2019 period, right? All this is a bit confusing. Therefore, I recommend to use a single reference period, which will likely be 1979-2017.

Line 78-85: If possible, I would ask you to cite some articles that describe this dataset (and the other datasets you used).

Lines 95-97: Why did not you use only ERA5 data?

Lines 99-100: This sentence is not crystal clear. Can you rewrite it?

Lines 102-103: How did you regrid all model outputs?

Lines 104-106: This paragraph belongs to the following sub-chapter (Methods), I guess.

Lines 105-106: Have you considered using Mann-Kendall?

Line 129: How did you calculate the anomalies? Here and in other instances.

Lines 194-196: Is this part really necessary? Is it not obvious?

Lines 251-253: “The increasing pattern is similar to the observations, but the value is lower than the observations (figure not shown)”. Can you quantify this and explore a bit more the implications? This is not a secondary aspect of your research.

Line 255: I feel that there are too many “obviously” here and in the paper in general. Please, try to reduce them.

Line 264: “The rainfall changes (Figure 7i) confirm this speculation”. You correctly define this as a speculation. Please, try to tone down the parts of your article where you discuss about this.

Lines 281-284: You surf a lot on the wave of speculation, here. As I wrote before, please be cautious and try to embrace the uncertainty issue.

Line 313: 2000s

Figures 2: In panel (e), “swe If time series(79-17)”. What “If” means in the title? Also, here you write “swe” while in other figures you write SWE. Please, try to be consistent.

Table 1: Is the spatial resolution expressed in km? If yes, I recommend adding this information to the table.

Author Response

Comments on “Reasons for east Siberia winter snow water equivalent increase in the recent decades” by Wang et al. submitted to Remote Sensing

General comments

I have now read the paper “Reasons for east Siberia winter snow water equivalent increase in the recent decades”. The authors present a paper that aims at analysing the reasons of the winter (DJF) SWE increase since the year 1979 in east Siberia. For this region, the authors also provide projections of future SWE changes. They found that significant SWE increases occurred in the period 2000-2017, mainly in November (followed by the other winter months). The authors attributed this occurrence to NDJF precipitation increases due to the atmospheric dynamic effect and horizontal advection of moisture. Finally, through climate simulations, under high forcing scenarios, the authors showed that precipitation will continue to increase until 2100, while after the 2060s the precipitation could shift to rainfall type.

In general, the paper has clear and interesting aims. In addition, the reported findings are of great interest for the scientific community. Finally, the theme fits well into the scope of the journal Remote Sensing. However, there are several aspects that need to be improved/clarified before recommending this paper for publication.

Reply: Thanks very much for your comments!

Main comments

DATA AND METHODS

Currently, the description of your study area is completely missing. I understand that it is the east Siberia, but please consider that not all readers (including myself) are well informed about the characteristics of the region. Therefore, my suggestion is to add a Study area section to the Data and Methods where you could provide geographical and climatic information. In addition, can you provide here more information about estimated monthly SWE means in your study region? I think it would be nice to have a very rough idea about the seasonal SWE evolution in the analysed region (maybe there are even some observational data…). In this section, I also recommend to add a figure (it will be the Figure 1) in which you show the location of your study area in the world (or in the Northern Hemisphere, if you prefer), its borders (even roughly, like you did in Fig. 1a), etc.

Reply: Following the suggestion, we added a figure showing the location of the study area and the monthly SWE evolution in east Siberia from October to the following May.

In response, we add section 2.3 (Study area) in Data and Methods to describe the location and climate characteristics of east Siberia, and introduce the climatological monthly mean SWE evolution. (Lines 144-154, and Figure 1)

Datasets

Although I am not an expert in using all the datasets you used in this paper, I feel that several bits of information are missing here. In addition, I like the idea to keep the section shosrt (as well as the entire paper), but I think this is too much.

(i) I would recommend to clearly explain why you chose some datasets and to do what. At the moment, all possible explanations are up to the reader, since all your datasets “pop-up” in a menu-style list without further explanations. Please, “take the reader in the hand” and try to explain why and how you are doing things; you may want to do this also by describing all used datasets (and variables) in a table.

Reply: Following the suggestion, we modified the description of the datasets and added a table for the resolution and web site of the datasets. (Lines 82-105, and Table 1)

(ii) In this study, you completely rely on gridded remote sensed and reanalysis data, which is not necessarily bad. However, have you considered validating some of the used products with (potentially) available observational data? Do you have some ideas about the estimated errors and uncertainties associated to the products you used in your study area? Please, try to be very precise and rigorous since this is necessary to allow the reader assessing the overall quality of your approach and the uncertainties in your results.

Reply: In this study, the analyses are mainly based on GlobSnow v3.0 SWE and ERA5 reanalysis. The SWE is generated by Bayesian assimilation with synoptic snow depth observations and spaceborne passive microwave brightness temperatures from satellites (Luojus et al. 2021). The SWE data were validated by in situ snow course SWE data by the data publisher. Snow course observations consist of manual gravimetric snow measurements made at multiple locations along pre-defined transects several hundreds of meters to several kilometers in length, which are averaged together to obtain a single SWE value for a given transect on a given date. The in situ snow course SWE data were obtained from Canada, Finland, and Russia. ERA5 is the newest generation ECMWF reanalysis dataset and it performs well in the high latitudes of the Northern Hemisphere (Barrett et al. 2020; Graham et al. 2019). To validate the results obtained from ERA5 data, we redo the related analysis using the NCEP2 data, and the results are consistent. However, due to the paper length limitation, we only show the results based on ERA5 reanalysis.

In response, we added the following sentences:

‘The dataset was constructed by combining satellite-based passive microwave radiometer data with ground-based synoptic snow depth observations on the Northern Hemisphere terrestrial (non-high mountainous areas) and validated by independent in situ SWE data’ (Lines 83-86)

ERA5 is the newest generation ECMWF reanalysis dataset and it performs well in the high latitudes of the Northern Hemisphere (Barrett et al. 2020; Graham et al. 2019)’ (Lines 93-95)

(iii) Have you assessed the uncertainties in your scenarios? Currently, I feel that too many “certainties” about your outcomes and predictions are conveyed by your paper. For instance, this is what you wrote in the abstract “As east Siberia is cold in winter, even under the high radiative forcing scenario, precipitation in east Siberia will continue to increase and is almost dominated by snowfall until the 2060s. Thereafter, the precipitation will shift to rainfall type…”. I suggest being more cautious (including your writing style), also considering that a shift from solid to liquid precipitation is not trivial to model and predict; great uncertainties might exist in this. You might want to explain how you quantified the uncertainties here or, better, in the following sub-chapter (Methods).

Reply: Indeed, model projections may have uncertainty. To make the results more reliable, we used 18 CMIP6 models to calculate the ensemble mean and compared the historical model simulation results with the observations. Figure 7 shows that the precipitation in the regions near the coast increased slightly or even decreased after 2000, but the increase of precipitation is obvious in most other regions over east Siberia and certainly larger after than before 2000. There is a difference in the magnitude between model results and observations, but the trend of this change is similar.

In response, we added sentences about the uncertainty of the model results and modified sentences about the reliability of the model results. (Lines 26-28, 110-112 305-310318, and 340)

RESULTS

The Results section is an amalgam of results, interpretation, discussion, and methodology. Lines 132-134: this is discussion. Lines 159-166, this part should be in the methodological section (it is relevant for your work). Are there previous studies that applied this approach? Can you provide some references? Lines 213-219: mixture of results and interpretation. Lines 238-240, 261-264, and 281-284: this is interpretation (further details below).

Reply: Following the suggestion, we rearranged the contents. (Lines 162, 187-189, 269, 295, and 319)

CONLUSIONS AND DISCUSSION

I think that this part should be completely rewritten.

(i) Currently, this section does not contain a discussion nor a conclusion. Essentially, you somehow repeat what you already wrote in the previous section (except the last paragraph). I suggest describing your results and interpreting them (you did it, already), second, you should discuss them in the context of previous research. Merging and rewriting the current Results section with the current Conclusions and Discussion might help you in doing this, obtaining a new section: Results and Discussion.

Reply: Following the suggestion, we modified the sentences in this section with removal of repetition of those descriptions in previous sections. (Lines 326-358)

(ii) At the moment, your paper heavily relies on the interpretation rather than discussion of data. I understand that this is not an extremely specific comment, but I would suggest you to try improving your discussion with this in your mind. For instance, are there other regions/area in the world with increasing SWE in the recent decades/years? Are there papers discussing about these regions/areas? Any analogies with your case? What are the implications of your findings? Do you have remaining questions, doubts, new perspectives, etc.? These are just some suggestions.

Reply: Following the suggestion, we compared the present work with the previous research, point out that the issues remind unclear, and discuss the potential implication of the finding. (Lines 344-358)

(iii) Please, write your conclusions in a separated section. Currently, I cannot find a real conclusion in your paper. The Conclusions should summarise the conducted research both with respect to its place among the published works and in terms of future perspectives. Also, what did we learn from your study? Why is your study relevant for the scientific community? Therefore, in order to better convey the main findings of the paper, it would be better to write few, brief conclusive highlights and then adding some perspectives.

Reply: Following the suggestion, we have modified the last section with a focus on the main findings of the analysis. Descriptions of those provided in previous sections have been removed to avoid repetition. (Lines 326-358)

(iv) In the last paragraph you discuss some uncertainties regarding your approach. First, I do not think that this paragraph should be at the end of your paper, in the Conclusions. The take-home message for the reader seems to be this one, which is a methodological issue, essentially. I think you have way more relevant take-home messages and potential future research perspectives to explore. Second, this part could go into the new Results and Discussion section, maybe in a sub-chapter called Research limitations and uncertainties, in which you might also want to discuss the uncertainties related to your research (please, see previous comments about this issue).

Reply: We have moved this paragraph of discussion of uncertainties to the main text at the end of section 3.1. (Lines 199-205)

FIGURES

I find the resolution of the figures to be pretty low, which could be due to an automatic downsampling performed by the submission system (I do not know this). If this is not the case, I warmly recommend improving the resolution of all figures.

Reply: It should be due to automatic downsampling. We will upload the original figures if the paper is accepted for publication.

Detailed comments

Keywords: (1) “Snow water equivalent” - I would use “SWE” since you already used “snow water equivalent” in the title. (2) “Sources and reasons of increase” - This sounds like a weird keyword. I would use other keywords such as “precipitation increase”, “climate change”, “snowfall increase”, etc. These are just some hints.

Reply: Following the suggestion, we modified the keywords ‘snow water equivalent’ and ‘Sources and reasons of increase’ to ‘SWE’ and ‘precipitation increase’. (Line 29)

Line 25: “…and is almost dominated by snowfall until the 2060s”. How can something be “almost” dominated by something else? Can you rephrase this?

Reply: In response, we deleted the word ‘almost’. (Line 26)

Line 26: “Thereafter, the precipitation will shift to rainfall type”. I would be more cautious; properly communicating scientific uncertainty is vital, especially when this is about climatic scenarios and potential precipitation phase changes. In addition, from what you write, one could think that all solid precipitation will shift to liquid, which is not the case. Please, try to be more precise since this is a very strong statement, which is not even supported by your results. Indeed, there might be an increase in rainfall, although snowfall is still predicted to be predominant in most areas.

Reply: Thanks for your suggestion. We changed the last sentence in the abstract as follows: ‘Thereafter, with the rainfall increase and the accelerated snowmelt due to rising temperature, precipitation will gradually shift to rainfall type and the SWE may turn to decrease.’. (Lines 26-28)

Line 31-32: “Snow cover is the fastest-changing natural surface feature in the Earth’s climate system…”. What do you mean? Maybe you meant cryosphere instead of Earth’s climate system?

Reply: Snow cover is one of the important factors in the climate system, especially in cold seasons or regions. As snow cover is very sensitive to changes in air temperature, it may melt soon after snowfall as temperature rises. So, snow cover has a fast-changing character. However, some factors in the Earth's climate system also change very fast, for instance, thinner sea ice at the edge in the warm season. In order to make the description more appropriate, we modified the sentence.

‘Snow cover is one of the fastest-changing natural surface features in the Earth’s climate system’. (Lines 33-34)

Line 36: “…local and downstream atmospheric circulation changes”. Are you talking about mountains, here?

Reply: Like a river, flooding upstream can have an impact downstream. Atmosphere as fluid, the changes in one region may be followed by weather and climate changes in other regions located downstream.

Lines 41-41: “For example, about 50% of runoff in western Himalayan watersheds comes from seasonal snowmelt [14]”. I do not think this sentence is pertinent to your study. It would be better to focus on geographical settings (and environmental processes) more relevant for your study region (and similar regions).

Reply: Following the suggestion, we replaced the example and changed the reference.

‘For example, the increase in discharge corresponded to a decrease of snow cover during the snowmelt periods in the Siberian watersheds’ (Lines 41-43 and reference 14)

Lines 65-66: “East Siberia is cold, especially in winter when the majority of precipitation falls as snow and snowmelt is very little in cold seasons”. How cold? Can you provide more climatic information? Here or in the Study area section.

Reply: In response, we added words to describe the climate in east Siberia.

‘with an average annual air temperature of about -10°C and the lowest air temperature can even reach -70°C in winter’ (Lines 67-68)

Lines 72-73: “This study documents the sources and reasons of the winter (DJF) SWE increase in east Siberia during 1979-2018”. Here, you state that the investigated period is 1979-2018. However, the actual period you analysed for SWE trends is 1979-2017, right? In addition, for precipitation (and other climatic data) it seems that you used the 1979-2019 period, right? All this is a bit confusing. Therefore, I recommend to use a single reference period, which will likely be 1979-2017.

Reply: The SWE data we used in this study spans the period of January 1979-May 2018. In this paper, we focused on clod season’s (November to February) SWE changes, therefore we only analyzed data from 1979 to 2017 winter (actually up to February 2018). To make the trend analysis at the end of the time period more reliable, when analyzing the sources and causes of SWE changes, we used atmospheric variables extended to 2019.

In response, we modified and added sentences to describe.

‘This study documents the sources and reasons for the winter (DJF) SWE increase in east Siberia during recent decades.’ (Lines 75-76)

We focused on clod season’s (November to February) SWE changes.’ (Lines 88-89)

‘To make the trend analysis at the end of time period more reliable, we used atmospheric variables extended to 2019.’ (Lines 139-140)

Line 78-85: If possible, I would ask you to cite some articles that describe this dataset (and the other datasets you used).

Reply: Done. (Lines 83-86, 93-95, and 96-98)

Lines 95-97: Why did not you use only ERA5 data?

Reply: ERA5 is the newest generation ECMWF reanalysis dataset. It performs well in the high latitudes of the northern hemisphere, including precipitation and air temperature. In this study, we verified the ERA5 results using the NCEP2 reanalysis data, and the results are consistent. Due to length limit of the paper, we didn’t show the results of NCEP2 reanalysis.

In response, we modified sentences in section 2 to describe the ERA5 data. ‘To analyze the sources and reasons of SWE changes, we used monthly cloud cover, radiation flux on single level and wind, vertical motion, and specific humidity on pressure levels from the ECMWF Reanalysis v5 (ERA5). ERA5 is the newest generation ECMWF reanalysis dataset and it performs well in the high latitudes of the Northern Hemisphere.’ (Lines 91-95)

Lines 99-100: This sentence is not crystal clear. Can you rewrite it?

Reply: In response, we modified the sentence as ‘We used 18 model outputs from the CMIP6 to evaluate the precipitation, snowfall, and rainfall projection.’. (Lines 105-106)

Lines 102-103: How did you regrid all model outputs?

Reply: In this study, we use bilinear interpolation method to regrid the model outputs using the tool of the Earth System Modeling Framework (ESMF). The ESMF is software for building and coupling weather, climate, and related models.

In response, we added words to clarify the method. (Lines 109-110)

Lines 104-106: This paragraph belongs to the following sub-chapter (Methods), I guess.

Reply: We have moved the paragraph on the methods. (Lines 132-137)

Lines 105-106: Have you considered using Mann-Kendall?

Reply: Following the suggestion, we redo the test using the Mann-Kendall test method and replaced the figures. (Lines 142-143; Figures 2-6, and 8)

Line 129: How did you calculate the anomalies? Here and in other instances.

Reply: In this study, the anomalies are calculated by the total values minus the climatology values in the time period.

In response, we added sentence to clarify. (Lines 140-141)

Lines 194-196: Is this part really necessary? Is it not obvious?

Reply: Deleted. (Lines 224)

Lines 251-253: “The increasing pattern is similar to the observations, but the value is lower than the observations (figure not shown)”. Can you quantify this and explore a bit more the implications? This is not a secondary aspect of your research.

Reply: Following the suggestion, we calculated the trend of historical precipitation over east Siberia for the period of 2000-2014. The value is 0.4 mm/month/10a.

In response, we changed the sentence as follows: ‘The increasing pattern is similar to the observations (figure not shown). The trend of area mean is 0.4 mm/month/10a for the period 2000-2014, lower than the observations’ (Lines 283-284)

Line 255: I feel that there are too many “obviously” here and in the paper in general. Please, try to reduce them.

Reply: In response, we removed them.

Line 264: “The rainfall changes (Figure 7i) confirm this speculation”. You correctly define this as a speculation. Please, try to tone down the parts of your article where you discuss about this.

Reply: We modified the sentences. (Lines 293-295)

Lines 281-284: You surf a lot on the wave of speculation, here. As I wrote before, please be cautious and try to embrace the uncertainty issue.

Reply: Following the suggestion, we modified the sentences to indicate the uncertainty of the projection results and moved the interpretation part to the conclusion section.

We modified the sentences as follows: ‘After the 2060s, the precipitation gradually shifts to the rainfall type.’ (Lines 318-319)

Line 313: 2000s

Reply: Corrected. (Line 332)

Figures 2: In panel (e), “swe If time series(79-17)”. What “If” means in the title? Also, here you write “swe” while in other figures you write SWE. Please, try to be consistent.

Reply: ‘swe lf time series (79-17)’ denote the time series of low-frequency SWE changes. Following the suggestion, we modified the title of figure 3(e) to ‘SWE time series (79-17)’. (Line 173, Figure 3e)

Table 1: Is the spatial resolution expressed in km? If yes, I recommend adding this information to the table.

Reply: In the previous vision, it is the number of grid points in longitude and latitude. In the present, we used the grid degree.

In response, we modified the table. (Line 115, table 2)

References:

Luojus, K., Pulliainen, J., Takala, M., Lemmetyinen, J., Mortimer, C., Derksen, C., ... & Venäläinen, P. GlobSnow v3. 0 Northern Hemisphere snow water equivalent dataset. Scientific Data, 2021; 8(1), 1-16.

Barrett, A. P., Stroeve, J. C., & Serreze, M. C. Arctic Ocean precipitation from atmospheric reanalyses and comparisons with North Pole drifting station records. J. Geophys. Res. Oceans, 2020, 125(1), e2019JC015415.

Graham, R. M., Hudson, S. R., & Maturilli, M. Improved performance of ERA5 in Arctic gateway relative to four global atmospheric reanalyses. Geophys. Res. Lett., 2019, 46(11), 6138-6147.

Round 2

Reviewer 2 Report

This is the second review of the paper: “Reasons for east Siberia winter snow water equivalent increase in the recent decades”.  Overall, this paper is pretty close.  I still think that the authors should do a little more to acknowledge or test the role of snowmelt vs transitioning from snowfall to rainfall in the future climate scenarios (in section 3).  At least sentence or two about this is warranted (not just in the abstract).  Other than that, the paper could still benefit from another proofreading as there are still some grammatical errors and phrasings that are awkward (some of which are point out below).

Line 21: “November SWE and followed by that in winter” - suggest that you rephrase this sentence

Line 46: Change “that driving” to “that drive”

Line 129: suggest changing the other “under 700 hPa”.  Is this correct that it is from 1000-700 hPa.  Is the surface at 1000 hPa?

Suggest putting section 2.3 (Study area) in front of 2.1 and 2.2 (i.e. making it the first section in section 2.

Line 151: What is this “remarkable annual variation”, and why is it important in the context of this paper?  Could it be shown on the figure somehow?

Line 188: Could you rephrase this passage (the trend has no obvious trend?)

Line 197-202: This passage is still extremely confusing.  Is it necessary to include?

Line 314: Please at least mention the impact of snow ablation here.  Snowfall changing to rainfall, although important, is not the whole story as SWE is the affected by both accumulation and ablation.  Ideally, the authors would check model quantities like SWE/Snowfall to get a sense for when mid-winter ablation starts to limit SWE, but at least include a sentence or two on this effect.

Author Response

This is the second review of the paper: “Reasons for east Siberia winter snow water equivalent increase in the recent decades”.  Overall, this paper is pretty close.  I still think that the authors should do a little more to acknowledge or test the role of snowmelt vs transitioning from snowfall to rainfall in the future climate scenarios (in section 3).  At least sentence or two about this is warranted (not just in the abstract).  Other than that, the paper could still benefit from another proofreading as there are still some grammatical errors and phrasings that are awkward (some of which are point out below).

Reply: Thanks very much for your comments! We replied point by point as follows.

Line 21: “November SWE and followed by that in winter” - suggest that you rephrase this sentence

Reply: Following the suggestion, we modified the sentence.

‘SWE in November, followed by that in winter’ (Line 21)

Line 46: Change “that driving” to “that drive”

Reply: Corrected. (Line 47)

Line 129: suggest changing the other “under 700 hPa”.  Is this correct that it is from 1000-700 hPa.  Is the surface at 1000 hPa?

Reply: In this study, the ERA5 wind and specific humidity of pressure levels were used. Indeed, the surface over east Siberia is not all around 1000 hPa. But most regions of east Siberia are under 1000 m (about 900 hPa). So the differences are small between the variables integrated from 1000 hPa to 700 hPa and integrated from surface to 700 hPa.

In response, we modified the other ‘under 700 hPa’ to ‘from 1000 hPa to 700 hPa’. (Lines 142, 234)

Suggest putting section 2.3 (Study area) in front of 2.1 and 2.2 (i.e. making it the first section in section 2.

Reply: Done. (Lines 81-91)

Line 151: What is this “remarkable annual variation”, and why is it important in the context of this paper?  Could it be shown on the figure somehow?

Reply: The annual variation refers to the month-to-month change in  mean SWE. In response, we modified the sentence.

‘The mean SWE has a remarkable month-to-month change.’ (Lines 89-90)

Line 188: Could you rephrase this passage (the trend has no obvious trend?)

Reply: We modified the sentence as: ‘In October, the trend of the SWE difference is sporadic and has no obvious spatial characteristics’ (Lines 192-193)

Line 197-202: This passage is still extremely confusing.  Is it necessary to include?

Reply: Following the suggestion, we deleted those sentences. (Line 201)

Line 314: Please at least mention the impact of snow ablation here.  Snowfall changing to rainfall, although important, is not the whole story as SWE is the affected by both accumulation and ablation. Ideally, the authors would check model quantities like SWE/Snowfall to get a sense for when mid-winter ablation starts to limit SWE, but at least include a sentence or two on this effect.

Reply: Convincing analytical results cannot be directly shown due to the lack of sufficient model-simulated snowmelt or SWE data. We modified the sentence to explain the role of snowmelt in the warming climate that may cause SWE to decrease after the 2060s over east Siberia. (Lines 314-316)

Reviewer 3 Report

The authors addressed all the comments I made and I believe that they had improved the manuscript. Before accepting it for publication, I have only three final suggestions:

(1) I would still recommend to help the reader in understanding the uncertainties in the future scenarios. Thus, I suggest adding three figures in the Supplementary Material. In each figure you could show all model outputs (whose average you used for creating Figure 9), for each variable: pre, sf, and rf. This will certainly help understanding the different behaviour of each model, for each variable.

(2) Please call chapter 4 "Discussion and conclusions", since it is rather weird to write the other way around.

(3) Please check again the manuscript for grammar mistakes. E.g., Lines 352-353 “Observational analysis and model simulation suggested that Siberia snow HAVE an influence on the Northern Hemisphere circulation…” - “HAS”.

Author Response

The authors addressed all the comments I made and I believe that they had improved the manuscript. Before accepting it for publication, I have only three final suggestions:

Reply: Thanks very much for your suggestions!

(1) I would still recommend to help the reader in understanding the uncertainties in the future scenarios. Thus, I suggest adding three figures in the Supplementary Material. In each figure you could show all model outputs (whose average you used for creating Figure 9), for each variable: pre, sf, and rf. This will certainly help understanding the different behaviour of each model, for each variable.

Reply: Following the suggestion, we have drawn the figures to show each model outputs time series of area-mean NDJF precipitation (pre), rainfall (rf), and snowfall (sf) over east Siberia. As the variables of each model projections under SSP585 scenarios have large uncertainties, we show each variable in a sub-figure to indicate the range of changes, rather than draw an individual sub-figure for each model of each variable.

In response, we added the figure in Appendix as Figure S1 and added a sentence referring to it. (Lines 316-318, Figure S1)

(2) Please call chapter 4 "Discussion and conclusions", since it is rather weird to write the other way around.

Reply: Done. (Line 324)

(3) Please check again the manuscript for grammar mistakes. E.g., Lines 352-353 “Observational analysis and model simulation suggested that Siberia snow HAVE an influence on the Northern Hemisphere circulation…” - “HAS”.

Reply: Checked. Corrected. (Line 357)
